


# Atmospheric observations consistent with reported decline in the UK's methane emissions, 2013 – 2020

Mark F. Lunt[1], Alistair J. Manning[2], Grant Allen[3], Tim Arnold[4,1], Stéphane J-B. Bauguitte[5], Hartmut Boesch[6], Anita L. Ganesan[7], Aoife Grant[8], Carole Helfter[9], Eiko Nemitz[9], Simon J. O'Doherty[8], Paul I. Palmer[1,10], Joseph R. Pitt[3,*], Chris Rennick[4], Daniel Say[8], Kieran M. Stanley[11], Ann R. Stavert[12], Dickon Young[8], and Matt Rigby[8]

[1]School of GeoSciences, University of Edinburgh, Edinburgh, UK
[2]Hadley Centre, UK Met Office, Exeter, UK
[3]Department of Earth and Environmental Sciences, University of Manchester, Manchester, UK
[4]National Physical Laboratory, Teddington, UK
[5]FAAM Airborne Laboratory, National Centre for Atmospheric Sciences, Building 146, College Road, Cranfield, UK
[6]National Centre for Earth Observation (NCEO), University of Leicester, Leicester, UK
[7]School of Geographical Sciences, University of Bristol, Bristol, UK
[8]School of Chemistry, University of Bristol, Bristol, UK
[9]Centre for Ecology and Hydrology, Penicuik , UK
[10]NCEO, University of Edinburgh, Edinburgh, UK
[11]Institute for Atmospheric and Environmental Science, Goethe University Frankfurt, Frankfurt, Germany
[12]CSIRO Oceans and Atmosphere, Aspendale, Victoria, Australia
[*]now at School of Marine and Atmospheric Sciences, Stony Brook University, Stony Brook, NY, USA

**Correspondence:** Mark Lunt (mark.lunt@ed.ac.uk)

**Abstract.**

Atmospheric measurements can be used as a tool to evaluate national greenhouse gas inventories through inverse modelling. Using eight years of continuous methane ($CH_4$) concentration data, this work assesses the United Kingdom's (UK) $CH_4$ emissions over the period 2013–2020. Using two different inversion methods, we find mean emissions of $2.10 \pm 0.09$ Tg yr$^{-1}$

and $2.12 \pm 0.26$ Tg yr$^{-1}$ between 2013–2020, and an overall trend of -0.05 $\pm$0.01 Tg yr$^{-2}$ and -0.06 $\pm$0.04 Tg yr$^{-2}$, a 2–3% decrease each year. This compares with the mean emissions of 2.23 Tg yr$^{-1}$ and trend of -0.03 Tg yr$^{-2}$ (1% annual decrease) reported in the UK's 2021 inventory between 2013–2019. We examine how sensitive these estimates are to various components of the inversion set-up, such as the measurement network configuration, the prior emissions estimate, the inversion method, and the atmospheric transport model used. We find the decreasing trend to be due primarily to a reduction of emissions from

England, which accounts for 70% of the UK $CH_4$ emissions. Comparisons during 2015 demonstrate consistency when different atmospheric transport models are used to map the relationship between sources and atmospheric observations at the aggregation level of the UK. The posterior annual national means and negative trend are found to be consistent across changes in network configuration. We show, using only two monitoring sites, the same conclusions on mean UK emissions and negative trend would be reached as using the full six-site network, albeit with larger posterior uncertainties. However, emissions estimates

from Scotland fail to converge on the same posterior under different inversion setups, highlighting a shortcoming of the current observation network in monitoring all of the UK. Although $CH_4$ emissions in 2020 are estimated to have declined relative





to previous years, this decrease is in line with the longer-term emissions trend, and is not necessarily a response to national lockdowns.

# 1 Introduction

The United Kingdom (UK) is one of many countries to have made a commitment to reduce their greenhouse gas (GHG) emissions, with the UK parliament creating a legally binding target of achieving net zero carbon emissions by 2050 under the Climate Change Act (2008) (UK Parliament, 2008). Each year the UK compiles a National Atmospheric Emissions Inventory (NAEI) for greenhouse gases (Brown et al., 2021), which forms the basis for the National Inventory Report that is submitted to the United Nations Framework Convention on Climate Change (UNFCCC). This report provides an annual stock-take of

emissions of all the gases covered under the Kyoto Protocol, from 1990 to two years preceding the current year. In the 2020 submission the total reported emissions of greenhouse gases from the UK for 2018 was 456 Tg $CO_2$-equivalent, down from the 1990 baseline value of 798 Tg $CO_2$-equivalent, a 43% reduction (Brown et al., 2020).

The annual inventory reports allow the progress towards the climate change act target to be tracked. These reports are compiled from a detailed collection of emission factors and activity data for each source sector. The uncertainties in these

data can be large for certain sectors or gases. As such, independent evaluation through atmospheric measurements can play an important role in targeting gases or source sectors for inventory improvement. Indeed, the UK's annual inventory report contains an annex which compares the reported values for each gas with values inferred from atmospheric measurements. However, although this is considered best practice (IPCC, 2006) there is currently no legal obligation for countries to do so.

Of the 456 Tg $CO_2$-equivalent reported for 2018, 369 Tg $CO_2$-equivalent (81%) was a result of $CO_2$ emissions, whilst 52 Tg

$CO_2$-equivalent (11%) was due to methane ($CH_4$). With a lifetime of 12.4 years (Myhre et al., 2013), $CH_4$ is considered to be a short-lived climate pollutant, the reduction of which could reduce short-term radiative forcing (Shindell et al., 2012). According to the NAEI 2020 report, the primary sources of UK anthropogenic $CH_4$ emissions of 2.08 Tg in 2018 were agriculture (1.02 Tg, 49%), waste (0.77 Tg, 37%) and energy production (0.28 Tg, 13%). The NAEI has an estimated 95% confidence range on these annual $CH_4$ emissions of 1.80–2.48 Tg yr$^{-1}$. In addition to anthropogenic sources, $CH_4$ is emitted naturally from

environments such as natural wetlands and as a product of biomass burning from wildfires. Whilst these sources are significant globally (Saunois et al., 2020), the vast majority of the UK's emissions are anthropogenic (Bergamaschi et al., 2010; Ganesan et al., 2015), potentially making the evaluation of the NAEI total though atmospheric measurements simpler than in countries where a more substantial fraction originates from natural sources.

Evaluating emissions using atmospheric measurements can be achieved through the process of inverse modelling. Over the

last decade there has been a move towards establishing dedicated national greenhouse gas monitoring networks for this purpose in some countries. In 2012, the Deriving Emissions linked to Climate Change (DECC) network was established in the UK, measuring GHG mole fractions at three sites in the UK, in addition to the long-running Mace Head station in Ireland (Stanley et al., 2018). Ganesan et al. (2015) used measurements from this network to estimate UK methane emissions of 2.09 (1.65–2.67) Tg yr$^{-1}$ from August 2012 – August 2014, which were in agreement with the NAEI. In Switzerland, the CarboCount-CH





network was established, consisting of four measurement sites, dedicated to measuring GHG fluxes at high spatial and temporal resolution (Oney et al., 2015). The network was used by Henne et al. (2016) for verification of the Swiss methane inventory, who found that their posterior estimate was largely in agreement with the Swiss national inventory report, but with reduced uncertainty. Pison et al. (2018) explored the constraint on $CH_4$ emissions from France in 2012, using data from four stations in France and five from the UK, Ireland and Netherlands. The study found that emissions could be resolved from regions

of about $5 \times 10^4$ km$^2$ and on a timescale of about one week. Although not all areas of France were well-constrained by the data, emissions on the annual timescale were estimated with an uncertainty of less than 10%. Similar multi-site measurement networks have been used in the US to estimate methane emissions from California (Jeong et al., 2013), where it was found that the primary source of uncertainty was an under-sampling of urban areas.

Regional scale inverse modelling studies rely on regional atmospheric transport models to map the relationship between

emissions and atmospheric mole fractions. Challenges in this process include accounting for the boundary conditions at the edge of the regional model domain, and accurate modelling of atmospheric transport at high temporal and spatial resolution, particularly vertical transport (Bergamaschi et al., 2018). Uncertainties associated with these issues can limit the useful information that can be derived from regional networks. For example, Bergamaschi et al. (2015) used a network of 10 continuous measurement sites across Europe to estimate national methane emissions from European countries, using a number of dif-

ferent transport models and inversion approaches. Their results found that significant differences occur in national estimates dependent upon the transport model or inversion method used, suggesting systematic differences may exist between models. Similarly, Brunner et al. (2017) found the national-scale outputs of four different inverse modelling systems used for hydrofluorocarbon emissions estimation often did not overlap within the stated analytical uncertainties, suggesting that unaccounted for systematic uncertainties are a significant contributor to posterior emissions uncertainty.

The Greenhouse gAs Uk And Global Emissions project (GAUGE) was conceived as a means of robustly constraining UK GHG emissions and to provide insight on the effectiveness of the UK's GHG reduction policies (Palmer et al., 2018). The project included additional tall tower sites at Bilsdale in northern England and Heathfield, south of London (Stavert et al., 2019). These two tower sites added to the existing DECC network infrastructure at the time which comprised of two sites in England, and one each in Scotland and Ireland, as described in Ganesan et al. (2015). The GAUGE project further included

measurements from a church tower in Cambridgeshire, as well as a mobile site on a ship of opportunity off the East coast of the UK (Helfter et al., 2019), and a set of aircraft flights. This work uses the data of the UK DECC network and the additional available data of the GAUGE project to evaluate UK $CH_4$ emissions. Specifically, we explore how robust our emissions estimates are to changes in various components of the inverse modelling framework. These components include the number of measurement sites, the inverse modelling method, the transport model and the prior estimate of emissions used. This

work seeks to determine the impact of the above components on the results, and the extent to which we can have confidence in our evaluation of the national $CH_4$ inventory.



**Table 1.** Location, inlets, instruments and sampling dates at each measurement site.

| Site | Location | Instrument | Inlet heights (m a.g.l.) | Dates used in this work |
|------|----------|------------|--------------------------|-------------------------|
| BSD | 54.359°N, -1.150°E | Picarro | 42, 108, 248 | Feb 2014–Dec 2020 |
| HFD | 50.977°N, 0.230°E | Picarro | 50, 100 | Jan 2014–Dec 2020 |
| MHD | 53.326°N, -9.904°E | GC-FID | 10 | Jan 2013–Dec 2020 |
| RGL | 51.997°N, -2.540°E | Picarro | 45, 90 | Jan 2013–Dec 2020 |
| TAC | 52.518°N, 1.139°E | Picarro | 54, 100, 185 | Jan 2013–Dec 2020 |
| TTA | 56.555°N, -2.986°E | Picarro | 222 | May 2013–Sep 2015 |
| GLA | 52.460°N, -0.304°E | FTIR | 25 | Mar 2015–Dec 2015 |
| Ferry | Various | Picarro | 20 | Mar 2014–Dec 2016 |
| Aircraft | Various | LGR FGGA | 100–3000 | 7 flights between May 2014–Sep 2014 |

## 2 Measurements

### 2.1 DECC Tower network

Methane observations for this study were taken from six tower sites across the UK and Ireland: Mace Head, Ireland (MHD),
Ridge Hill, England (RGL), Tacolneston, England (TAC), Angus, Scotland (TTA), Bilsdale, England (BSD) and Heathfield,
England (HFD). The locations of the measurement sites, instrumentation and time period covered are given in Table 1.

With the exception of MHD, $CH_4$ mole fractions were measured at each tall tower site using a Picarro Cavity Ring-Down
Spectroscopy (CRDS) instrument. These measurements were calibrated using dry air standards in aluminium cylinders on the
WMO-2004A scale. Methane measurements at MHD were made using a gas chromatograph and flame ionization detector
(GC-FID) every 40 minutes. Calibration of the GC-FID measurements on the Tohoku university scale was performed using
standards filled wet in electropolished stainless steel cylinders (Ganesan et al., 2015; Prinn et al., 2018). To ensure consistency
between the two calibration scales, observations from the five sites calibrated on the WMO-2004A scale were multiplied by a
factor of 1.0003 (Dlugokencky et al., 2005). For tower sites with more than one inlet (BSD, HFD, RGL, TAC), measurements
from the highest inlet were used in this study, in an attempt to reduce the impact of local influences on the posterior emission
estimates.

$CH_4$ data from an additional short-term measurement site in Glatton, Cambridgeshire were also used. These measurements
were made on top a 25 m high church tower using a Fourier Transform Infrared Spectroscopy (FTIR) instrument and calibrated
on the WMO-2004A scale. Information on the instrumentation, inlet heights and data availability from each site is detailed in
Table 1.

### 2.2 Ship-based measurements

As part of the GAUGE project, $CH_4$ measurements were taken on-board a DFDS Seaways commercial freight ferry serving a
route between Rosyth, Scotland and Zeebrugge, Belgium. Return journeys were completed three times a week with the ship





charting a course just off the east coast of the UK for much of its journey, providing a regular transect of England and southern Scotland (Figure 1). Measurements from the Zeebrugge–Rosyth ferry were made on a Picarro CRDS system, and calibrated on the WMO-2004A calibration scale. Measurements were taken from the bow of the ship. Further details on the measurement setup and calibration are given in Helfter et al. (2019). Data from times when the ferry was in, or within, 50 km of either port were discarded due to the likely proximity of anthropogenic sources which would be unresolved at the resolution of the transport model output.

### 2.3 Aircraft

$CH_4$ measurements were taken onboard the UK's FAAM (Facility for Airborne Atmospheric Measurements) BAe-146 research aircraft, using a Los Gatos Fast greenhouse gas analyser (FGGA) instrument (O'Shea et al., 2013; Pitt et al., 2019). Continuous $CH_4$ measurements were made in conjunction with altitude, longitude and latitude coordinates. Measurements were calibrated on the WMO-2004A calibration scale, with calibrations performed on an hourly basis during flights. As part of the GAUGE project, a number of flights were conducted with a variety of flight paths including orbits of the British Isles to more dense trajectories upwind and downwind of London (Palmer et al., 2018). In this study we used data from seven of these flights over the course of four different months, between May and September 2014 (see Fig. 1).

## 3 Atmospheric models

### 3.1 NAME

The Numerical Atmospheric dispersion Modelling Environment (NAME, Jones et al., 2007; Manning et al., 2011) was used to calculate the relationship between the emissions field and the mole fractions. The set-up for calculating this relationship at each of the sites followed that described in Manning et al. (2011) and Lunt et al. (2016). Model particles were released from each inlet height $\pm$ 20 m in the model and tracked backwards in time for 30 days. The integrated residence time of the particles in the layer adjacent to the surface (0 to 40 m agl) was output on a lat-lon grid to give a direct measure of the sensitivity of mole fractions to changes in surface emissions from each grid cell. This grid has a resolution of $0.234° \times 0.352°$, equating to an approximate 25 km resolution. Annual mean (2015) NAME sensitivity footprints from each of the measurement sites are shown in Figure 1.



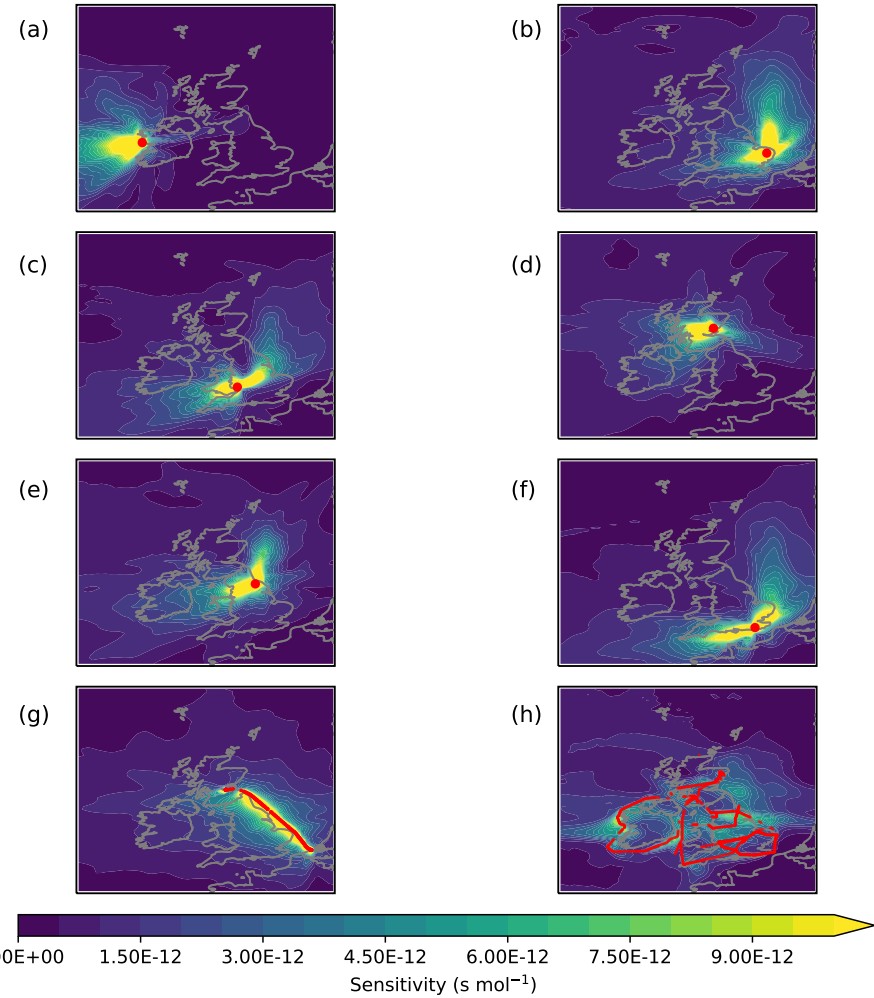

**Figure 1.** Annual mean NAME footprints for each of the measurement sites in 2015, showing the areas the measurements are most sensitive to (overall sensitivity extends much wider). (a) MHD, (b) TAC, (c) RGL, (d) TTA, (e) BSD, (f) HFD, (g) Ferry, (h) FAAM aircraft. Data for the FAAM aircraft (seven flights) are from 2014. Red dots show the measurement locations.

NAME was driven by offline meteorology fields from the UK Met Office's Unified Model (UM Cullen, 1993). The simulations used meteorology from a UK-specific mesoscale product at 1.5 km horizontal resolution (UKV) and one hour temporal resolution, nested within the model's global fields (UMG). The vertical structure of the mesoscale fields contained 57 levels up to a maximum height of 12 km, with 16 levels resolved in the lowest 1 km. The mesoscale meteorology was nested within the model's global meteorology fields, which were at an approximate horizontal resolution of 25 km up to July 2014, then 17 km until July 2017 and 12 km thereafter. The global fields contained 59 vertical levels up to a maximum height of 29 km, with approximately 11 levels in the lowest 1 km. The temporal resolution of the global fields were 3-hourly.





The UKV fields only cover a latitudinal-longitudinal area not much bigger than the UK. As a result of this, and as a conse-
quence of the larger computational burden of running at high resolution a number of changes were made to the default NAME
set-up as described in Lunt et al. (2016). The UKV meteorology was used to drive the transport of the particles within the
UKV area for the first 3 days after their release. Once particles left the UKV area further transport was dictated by the UMG
meteorology, and similarly after 3 days only the UMG meteorology was then available to transport the particles regardless of
location. The temporal dependence was applied to make the runs more computationally efficient, and sensitivity tests showed
there was no significant difference between having the UKV fields available for the full 30 day back-trajectory or only the first
3 days (since most particles will leave the limited UKV area within the 3 day period). Although NAME was run at the higher
resolution of the UKV meteorology, the footprints were still output on the same $0.234° × 0.352°$ grid, to ensure a regular grid
structure throughout the domain.

### 3.2  GEOS-Chem

A second set of model simulations were performed for 2015 using the GEOS-Chem model (Turner et al., 2015). The model
was run in a nested configuration, driven by meteorology from the GEOS-FP fields at $0.25° × 0.3125°$. The nested European
domain covered $40°N – 62°N$ and $15°W$ to $15°E$. Boundary conditions to this European domain were informed by a consistent
global simulation of the model at $2° × 2.5°$. The global simulation was driven by prior emissions from EDGAR v4.3.2 for
anthropogenic sources, the WetCHARTs v1.0 database for wetlands (Bloom et al., 2017) and GFED v4s for biomass burning
(van der Werf et al., 2017). The nested run used the same prior emissions field as for the NAME inversions, with the UK
NAEI distribution for the UK and EDGAR in all other countries. Sensitivities of the atmospheric measurements to emissions
were calculated using GEOS-Chem from 26 basis function regions in the European domain, with 14 of these for the UK and
Ireland. Sensitivities were also calculated to the magnitude of the global boundary condition fields at each of the four domain
boundaries, and the uncertainty in these fields explored in the inversion.

### 4  Prior emissions

To ensure that the prior assumptions did not influence our derived emissions trend, our prior emissions did not vary with
time. The a priori emissions spatial distribution was combined from several different sources for the inversions in this study.
Emissions from the UK (and surrounding ocean) were distributed according to the spatial distribution given by the UK's NAEI
from 2015 (NAEI2015, available at http://naei.beis.gov.uk/data/, last access:21/03/2021). These maps are provided for total
anthropogenic emissions as well as individual source sectors on an approximate $1 × 1$ km resolution grid. Outside of the UK,
emissions were distributed according to the EDGAR v4.3.2 inventory, which provides the emissions distribution at $0.1° × 0.1°$
resolution, regridded to the resolution of the NAME output. Figure 2 shows the spatial distribution of the NAEI2015 prior,
along with the breakdown of the three main NAEI2015 source sectors in the UK; agriculture, waste and energy. Emissions from
the agriculture sector are primarily concentrated in the western parts of the country, with emissions from livestock being the
main source. Waste and energy emissions are more concentrated around urban areas. In addition to the anthropogenic sources





there is likely a small natural component of methane emissions from the UK, associated primarily with methanogenesis in peatlands. These natural emissions were not accounted for in the NAEI2015 prior, and owing to the uncertainty over the exact spatial distribution, magnitude and temporal variation of these natural emissions, we ignore this relatively minor component of emissions in our prior emissions estimate. However, the main use of peatlands in the UK is for livestock grazing, and thus the areas where these emissions are expected to emanate are already accounted for in the spatial distribution of the prior. The inversion setup allows for the emissions from these regions to change in the inversion, if required. We acknowledge that the lack of a natural emissions component may lead to our estimates of anthropogenic UK methane emissions being slightly over-estimated, but explore the impact of the spatial distribution of prior emissions in section 6.5.

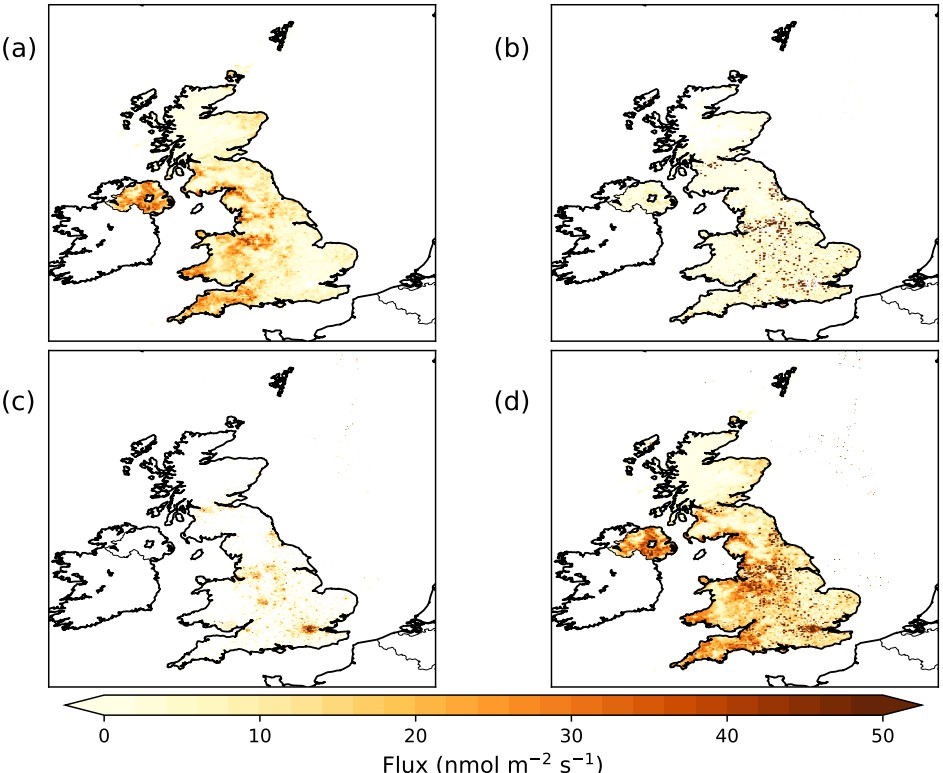

**Figure 2.** Maps showing the spatial distribution of the NAEI emissions in the UK at 5×5 km resolution, showing: (a) agriculture sector, (b) waste sector, (c) energy sector and (d) the total anthropogenic source.

## 5 Inversion Methodology

Inversions were performed using two separate methodologies. Unless stated otherwise, the majority of inversions were performed using the reversible-jump Markov chain Monte Carlo (rj-mcmc) method, described in Lunt et al. (2016), and briefly





summarised below. A further set of inversions were performed using the Inversion Technique for Emission Modelling (InTEM), which is the UK Met Office's inversion modelling system, and described in section 5.2.

## 5.1 Reversible-jump MCMC (rj-mcmc)

The rj-mcmc method (Green, 1995) is an extension of traditional Metropolis-Hastings MCMC (Metropolis et al., 1953; Hastings, 1970). In addition to exploring the probability density functions (PDFs) of the parameters (and hyper-parameters describing those PDFs), the number of unknowns is itself treated as an unknown. The method can take the uncertainty inherent in the aggregation of basis functions into account, providing a more robust estimate of the posterior parameters PDF. The rj-mcmc method can be applied to basis functions in one and two-dimensional problems in an effort to avoid making restrictive as-

sumptions about the discretization of parameter space (Sambridge et al., 2013). Starting from some prior PDF of parameters, the MCMC algorithm works by sampling from the target distribution of each parameter or hyper-parameter after it has been informed by the data. The PDFs are explored by perturbing the current state of the parameters model, $\mathbf{m}$, at each step of the chain to a new state, $\mathbf{m}'$. In the rj-mcmc algorithm this perturbation is chosen from one of the following proposals:

1. Update the parameters vector and prior hyper-parameters

2. Add a new region (birth)

3. Remove a region (death)

4. Move a region (move)

5. Update a model-measurement covariance hyper-parameter.

If the perturbation is favourable then the parameters model will move to the new state, $\mathbf{m}'$, or otherwise will remain un-

195 changed. Whether a proposal is favourable or not depends on a combination of the ratio of prior, proposal and likelihood probabilities of the current and proposed model. The prior probability, $\rho(\mathbf{m})$ describes how likely a particular model state is based on the form of the a priori PDF, and the ratio $\frac{\rho(\mathbf{m}')}{\rho(\mathbf{m})}$ gives the relative prior probabilities of the new and current model state. The likelihood ratio, $\frac{\rho(\mathbf{y}'|\mathbf{m}')}{\rho(\mathbf{y}|\mathbf{m})}$, relates the relative probabilities of predicting the data, given the new and current model states. The proposal ratio, $\frac{q(\mathbf{m}|\mathbf{m}')}{q(\mathbf{m}'|\mathbf{m})}$, describes the probability of picking the new model state from the current one, and vice

versa. The proposal is accepted provided the following equation is satisfied:

$$U \leq \left( \frac{\rho(\mathbf{m}')}{\rho(\mathbf{m})} \times \frac{\rho(\mathbf{y}'|\mathbf{m}')}{\rho(\mathbf{y}|\mathbf{m})} \times \frac{q(\mathbf{m}|\mathbf{m}')}{q(\mathbf{m}'|\mathbf{m})} \right), \tag{1}$$

where $U$ is a uniformly distributed random number between 0 and 1. Strictly speaking there is an additional term, a Jacobian matrix, $\mathbf{J}$ that should be taken into account for the dimension changing proposals. However, in the birth-death approach used here, where the new dimension is one less or one more than the current state, this term is always 1, and so can be ignored.

The acceptance criteria helps to ensure efficiency in sampling the target distribution, since unfavourable regions are rejected. Through exploring many thousands of samples, an estimate of the posterior distribution is reached.



The rj-mcmc algorithm was run for 200,000 iterations after a burn-in period of 50,000 iterations, and the chain thinned to store every 100th iteration giving 2000 stored samples of the posterior distribution. During the burn-in, tuning of the jump sizes for the parameters and hyper-parameter proposals was performed in order to achieve acceptance ratios of between 20–50% to ensure efficient exploration of the chain (Tarantola, 2005).

Unlike the parameter and hyper-parameter proposals there is no natural analogue of the jump size for the birth, death and move steps, since the acceptance ratio is more heavily dependent on the location of the proposed change as opposed to the value. However, whilst reasonable acceptance ratios were reported in Lunt et al. (2016), these were primarily due to the outer regions of the domain being relatively unconstrained by the data, leading to higher overall acceptance ratios. By contrast, birth, death and move proposals in regions of high sensitivity had lower acceptance ratios. To improve the efficiency of the algorithm the upper limit for the number of regions was set to 150 as a means of limiting the amount of time spent exploring regions in poorly constrained parts of the domain.

The rj-mcmc methodology was applied following the methodology in Lunt et al. (2016), where the basis functions each described a 2D spatial region, with an unknown number of these 2D spatial regions. The PDF for the number and location of spatial regions was set to be uniform and allowed to vary between 5 and 150, with a starting number of 50. Independent two-monthly inversions were performed with emissions assumed constant over each two month period.

In addition to the contribution from emissions, the $CH_4$ mole fractions are comprised of the underlying background variations. To estimate this background contribution in the rj-mcmc inversion, information on where the NAME particles left the NAME domain was stored to give an estimate of the sensitivity of the measurements to the mole fractions at the boundaries of the domain. These sensitivities were then combined with a climatology of mole fraction "curtains" from the global Eulerian Model for Ozone and Related Tracers (MOZART, Emmons et al., 2010) to give an estimate of the baseline mole fractions at each site. The MOZART mole fractions were generated using gridded emissions estimates from various methane sources including anthropogenic emissions from the Emissions Database for Global Atmospheric Research (EDGAR v4.3.2, EC-JRC/PBL, 2011), biomass burning (van der Werf et al., 2017), natural wetlands (Bloom et al., 2012) and other sources (Fung et al., 1991) as described in Lunt et al. (2016). These unconstrained MOZART generated curtains provided a prior estimate of the boundary condition mole fractions which were updated alongside emissions in the inversion. Optimized posterior mean model estimates of these baseline mole fractions are included in the supplement.

Using a hierarchical Bayesian approach (described in Ganesan et al., 2014), uncertainty parameters were themselves estimated in the inversion with the prior emissions uncertainty, model uncertainty and model error correlation time scale each described by a PDF. The prior uncertainty on each spatial emissions basis function was set to 50%, but described by a log-normal PDF which itself had a standard deviation of 50%, which was explored in the inversion. The prior model uncertainty was described by a gaussian PDF which had a mean of 20 ppb and standard deviation of 8 ppb. Different values for model uncertainty were estimated for each site every seven days. The correlation length scale, relating the covariance in time between measurement errors was fixed at 6 hours. The inversion using sensitivities calculated using GEOS-Chem followed a similar hierarchical MCMC approach. However, due to the computational complexity of calculating grid cell sensitivities with this





Eulerian model, these inversions used a fixed basis function definition and thus did not include the reversible-jump component of births, deaths and move proposals.

## 5.2 InTEM

The Inversion Technique for Emission Modelling (InTEM Manning et al., 2011; Arnold et al., 2018) has been developed over
many years and is the model used annually to estimate UK emissions of greenhouse gases in the UK national inventory report (Brown et al., 2020) submitted to the UNFCCC. InTEM is a Bayesian inversion system and assumes all errors are Gaussian but uses a non-negative least squares solver (Lawson and Hanson, 1974) preventing any negative solutions from being found.

InTEM uses the same NAME sensitivity footprints as used in the rj-mcmc inversions. The prior information comes from two sources: the first is the spatial distribution of $CH_4$ emissions as used in the rj-mcmc, namely UK NAEI2015 nested inside
EDGAR FT2010. The second is from an estimate of the time-varying boundary mole fractions of methane. The latter is derived from $CH_4$ observations at Mace Head during times identified as being representative of the well-mixed Northern Hemisphere baseline, namely times when the air has travelled predominately from Northern Canada with low influence from populated regions, high altitudes, local sources or southerly latitudes. A $4^{th}$-order polynominal is fitted to these data over a rolling 6-month window and a daily baseline uncertainty is estimated based on the root-mean-square of the fit.

InTEM allows the prior baseline to be adjusted based on 11 directional and height values depending on where the air enters the NAME modelled domain as described in Arnold et al. (2018). The inversion also allows for a site-specific bias adjustment to be made. The geographical grid used in the inversion is dependent on the sensitivity of the area to the observations, the higher the sensitivity the higher the resolution of the grid up to the native resolution of the NAME output as described in Manning et al. (2011), and the magnitude of the prior emissions from each area. In addition, the size of each grid area is limited to pre-defined
country boundaries. The model-observation uncertainty applied to the data varies for each 4-hour period and is comprised of three elements; observational uncertainty, baseline uncertainty and a meterological uncertainty. Observational uncertainty is estimated from the variability of the observations in the 4-hour period and baseline uncertainty is discussed above. The third element of uncertainty is proportional to the magnitude of the pollution event (10%), with a minimum uncertainty defined as the annual median $CH_4$ pollution event for each site (Manning et al., 2021).

## 5.3 Observation selection

Inverse methodologies generally assume that all observational and modelling errors are unbiased and random. Such a situation may not occur at certain times of the day, or under particular meteorological conditions, such as under particularly stable boundary layers where small errors in vertical mixing parameterizations could lead to significant errors in atmospheric trace gas distributions. A common approach to negate potential model biases is to use only observations from the middle of the day
(e.g. Peters et al., 2010; Bergamaschi et al., 2015). However, this approach has the disadvantage of ignoring the majority of the available data, and there is no guarantee that both the real and model atmosphere are similarly well-mixed. Indeed, this well-mixed criterion may only be appropriate during summer when the boundary layer stability tends to exhibit a pronounced





diurnal cycle. However it may not be met in the afternoon during winter months in particularly stable conditions. Conversely, potentially well-modelled data points may be excluded if they fall outside of this acceptable time window.

An alternative approach is to filter the data based on meteorological considerations. In this approach the atmospheric transport model is assumed to perform poorly under certain conditions such as stable boundary layers and low wind speeds, where unresolved sub-grid-scale processes (e.g. sea-breezes) or parameterized processes (e.g. convection) may dominate in reality but not in the model.

    We followed this approach through the use of a number of data filters, which differed slightly between InTEM and rj-mcmc

inversions. In the rj-mcmc inversions, for sites with more than one inlet height (BSD, HFD, RGL, TAC) we only used data from times when the difference between mole fractions recorded at different heights within the same hour was less than 10 ppb. This threshold was set to attempt to limit data to those times when the air was well-mixed. Since multiple inlets were not available at all sites, we further limited data use to times when the NAME simulated boundary layer height was greater than the measurement inlet height plus 250 m, and the local contribution of the 25 grid cells surrounding each measurement site to the

NAME footprint was smaller than 10% of the total footprint. These three filters were designed to maximise the probability of only including well-mixed air masses in our analysis and resulted in approximately 40–60% of available data being discarded for the rj-mcmc inversions, depending on the measurement site. InTEM inversions followed a similar setup but used a threshold of 20 ppb for the difference in $CH_4$ mole fraction between different inlet heights, and a fixed boundary layer height limit of 300 m.

## 290  6   Results

### 6.1   UK and Ireland emissions estimates 2013–2020

UK and Ireland $CH_4$ emissions are presented for the eight year period Jan 2013 to Dec 2020 inclusive, from the rj-mcmc and the INTEM inversions. Independent two-monthly inversions were performed using data from all available tall-tower sites and the MHD baseline station in each month. This varied from only three sites in early 2013 to all six sites in 2014 and 2015,

and five sites during 2016–2020. All uncertainties from the rj-mcmc inversions represent the 95% confidence interval which is given by the 2.5th and 97.5th percentiles of the posterior distribution. Uncertainties from the InTEM inversions represent $2\sigma$ standard deviations from the mean.





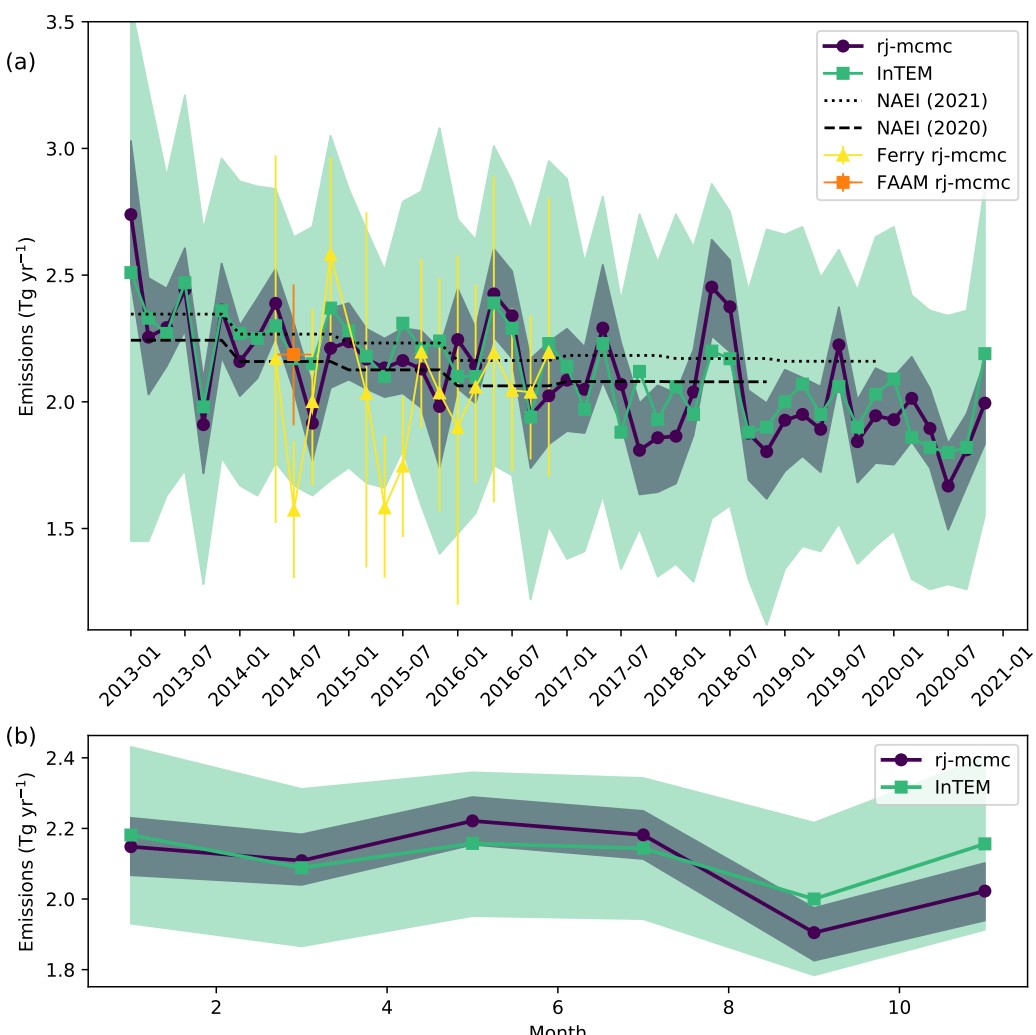

**Figure 3.** (a) Time series of two-monthly UK emissions from the two inversion methods, showing the rj-mcmc spatially varying inversion (purple) and InTEM inversion (green). rj-mcmc results are also included from the mobile measurement platforms of the ferry (yellow) and the FAAM aircraft (blue). The solid lines and markers designate the means, while the shading and uncertainty bars show the 95% confidence intervals of the posterior distributions. Black dashes show the NAEI annual mean from the two most recent reports (NAEI2020 and NAEI2021). (b) The mean seasonal cycle of two-month mean emissions from the full network inversions between 2013 and 2020.

Figure 3 shows the time series of derived posterior UK $CH_4$ emissions from the two different inversion set-ups. The two methods result in similar estimates of UK emissions, with a mean annual UK estimate from the rj-mcmc spatial inversion of 2.10 (2.01–2.18) Tg yr$^{-1}$ and 2.12 (1.86–2.38) Tg yr$^{-1}$ from InTEM. Due to the different treatment of model-measurement uncertainties in the inversions, the InTEM posterior emissions estimates have much larger confidence intervals. Over the eight-year inversion period 2013–2020 the annual emissions estimates from rj-mcmc show a negative trend of $-0.05 \pm 0.01$ Tg yr$^{-2}$.




InTEM results have a similar negative trend of $-0.06 \pm 0.04$ Tg yr$^{-2}$. This is equivalent to a 2% decrease each year, and broadly consistent with the estimated decrease in the NAEI of $-0.03$ Tg yr$^{-2}$.

Emissions estimates for 2020 were 1.89 (1.81–1.97) Tg yr$^{-1}$ and 1.93 (1.70–2.16) Tg yr$^{-1}$ from the rj-mcmc and InTEM inversions respectively. These estimates are in line with the trend in emissions from previous years, and do not indicate any substantial response to the national lockdowns enforced as a result of the coronavirus pandemic. Given the dominance of agriculture and waste sectors to UK CH$_4$ emissions, which are unlikely to have been impacted by the events of 2020, this finding is not unexpected.

The estimated mean emissions between 2013–2018 from both inversions of 2.16 (2.07–2.24) Tg yr$^{-1}$ and 2.17 (1.91–2.44) Tg yr$^{-1}$, rj-mcmc and InTEM respectively, are similar to the NAEI2020 reported emissions of 2.13 Tg yr$^{-1}$. However, in 2021 the inventory was revised to include a larger contribution of emissions from the LULUCF sector. These are estimated to contribute 0.19 Tg yr$^{-1}$ to the UK total emissions and largely account for the difference between the NAEI2021 and NAEI2020 reported emissions shown in Figure 3. The revised NAEI2021 emissions have a mean of 2.23 Tg yr$^{-1}$ between 2013–2019,

with a near constant offset compared to the previous year's submission.

It should be noted that the prior emissions used in the inversion (NAEI2015) ignored any potential contribution from natural emissions to the UK total, in common with previous versions of the NAEI. Within the additional LULUCF emissions added in the 2021 inventory, the majority of CH$_4$ emissions are split between grasslands (50%) and wetlands (42%). This type of land is mostly used for stock grazing (Brown et al., 2020) and as such it is likely that any wetland emissions would be spatially

indistinguishable from agricultural emissions in those parts of the country where these land types are prevalent. Thus, while natural emissions are not explicitly accounted for in the prior, it is unlikely that significant emissions would occur in areas where the prior does not already account for agricultural emissions. The sensitivity to the prior emissions distribution is explored in Section 6.5.

The bottom panel of Figure 3 shows the seasonal cycle of UK emissions derived from both inversions. The InTEM inversion

results do not show a large seasonal cycle, whilst the rj-mcmc results find the highest emissions on average during May–August. This is consistent with the period of highest surface temperature in the UK. The results shown in Fig. 3 represent the total methane emissions from the UK, and do not distinguish between different emissions sources. We divide these total CH$_4$ emissions into the respective sector contributions by assuming that the relative proportion of the NAEI sector split within each grid cell is correct but that the total magnitude is uncertain. Using the spatial distribution of the posterior estimates from

the rj-mcmc inversion we find that the summertime peak is most likely due to emissions from the agriculture sector. Posterior agriculture emissions during May–August are 0.14 Tg yr$^{-1}$ greater than other times of year, compared to 0.05 Tg yr$^{-1}$ greater emissions from waste and 0.01 Tg yr$^{-1}$ smaller emissions in the energy sector. This finding is qualitatively similar to that of Pison et al. (2018), who estimated a similar summertime peak in agriculture emissions from France.

As noted above, the overlap in spatial distributions of agricultural and LULUCF emissions mean that the estimated seasonal

cycle from the agriculture sector could instead reflect changes in grassland or wetland emissions. Indeed, the presence of a summertime peak in European CH$_4$ emissions estimates has previously been interpreted as evidence for the role of natural wetland CH$_4$ emissions across Europe (Bergamaschi et al., 2018). Whilst the most recent version of the NAEI (2021) explicitly





accounts for grassland and wetland emissions under the LULUCF category, spatial mapping of this distribution is not currently available preventing a direct comparison. There is a notable positive emissions anomaly in the rj-mcmc estimates (and to a

lesser extent InTEM) during the summer of 2018. In June-August 2018 the UK average temperatures were 1.4 °C above the 1981–2010 seasonal average and rainfall was 73% of the long-term seasonal average (data from https://www.metoffice.gov.uk/research/climate/maps-and-data/summaries/index., last access: 28/06/21).

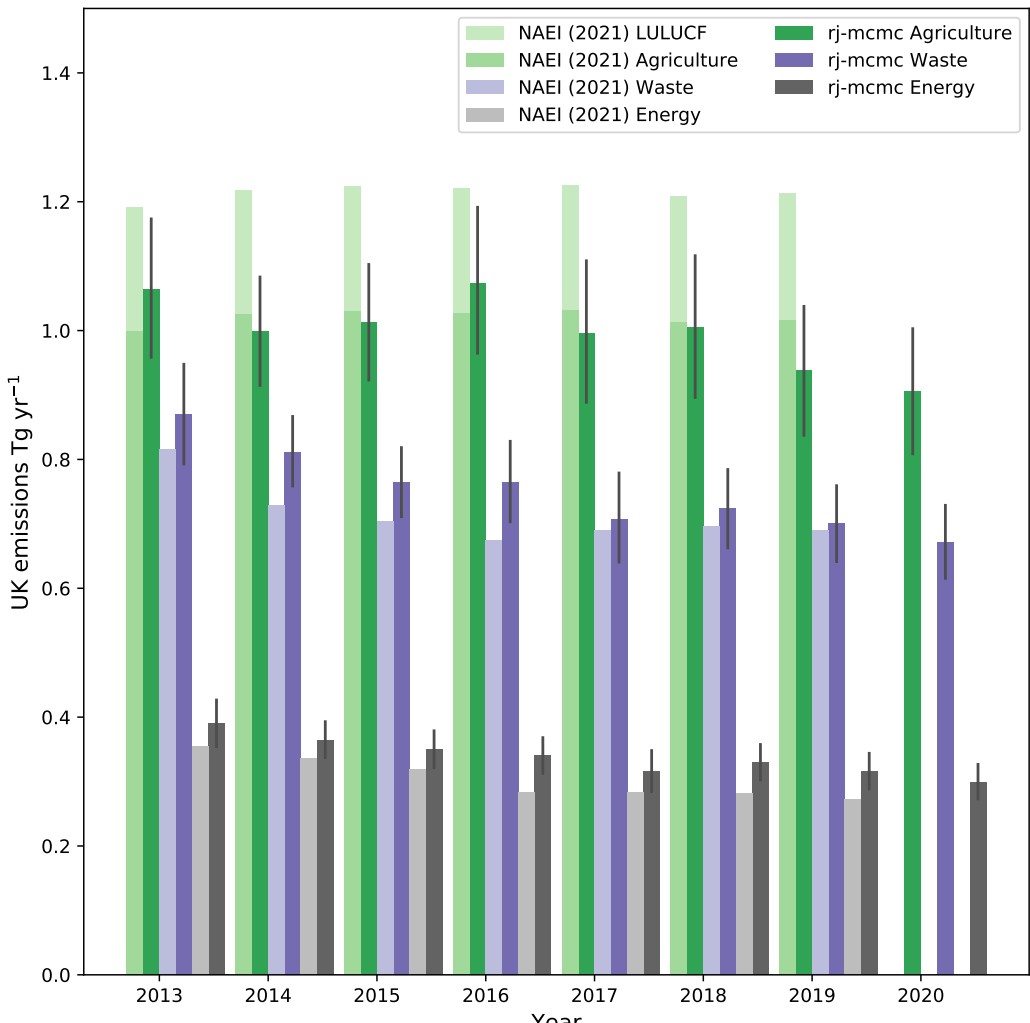

**Figure 4.** Annual UK emission estimates for agriculture and LULUCF (green), waste (purple) and energy (grey) sectors. The lighter shade in each colour represents the NAEI value, while the darker shade shows the average posterior distribution from the rj-mcmc inversion. Uncertainty bars represent the 95% confidence interval. Sector totals are calculated by scaling the individual prior sector distributions by the total posterior scale factor of each grid box. The rj-mcmc sector breakdown does not include an explicit representation of LULUCF.





Figure 4 shows the annual mean sector emissions from the rj-mcmc inversion, alongside the annual estimates from the 2021 inventory report. In common with the NAEI (2021), the agriculture section was found to have the largest emissions, with a mean over the 2013–2019 period of 1.01 (0.91–1.11) Tg yr$^{-1}$, compared to the NAEI average of 1.02 Tg yr$^{-1}$. The waste estimates were similar to the NAEI average of 0.71 Tg yr$^{-1}$ at 0.76 (0.69–0.82) Tg yr$^{-1}$. Emissions from the energy and industrial processes sectors were estimated to be 0.34 (0.31–0.37) Tg yr$^{-1}$ compared to the NAEI mean of 0.30 Tg yr$^{-1}$.

The largest component of the negative trend in the rj-mcmc UK total emissions are from the waste and energy sectors, at -0.05±0.01 Tg yr$^{-2}$ and -0.02±0.01 Tg yr$^{-2}$, respectively. Agriculture does not display a significant trend, in common with the NAEI. We stress that the NAEI trends are not included in the prior emissions used in the inversions, so the trends found in the posterior emissions estimates are independent of the NAEI reported trends. Pearson correlation coefficients between the sectors from the inversions are 0.6 between agriculture and waste, 0.4 between agriculture and energy and 0.7 between waste and energy. These correlation coefficients indicate some influence of changes in one distribution on any other. This may be due to overlap in the spatial distribution of each sector and the natural parsimony of the Bayesian solution, which favours broad-scale regional changes over finer resolution updates.

Posterior annual mean emissions estimates for Ireland from the rj-mcmc inversion are shown in Figure 5 and averaged 0.66 (0.61–0.72) Tg yr$^{-1}$ between 2013–2020. This compares to the Ireland's national inventory report average of 0.56 Tg yr$^{-1}$. We do not find any substantial trend in the annual mean rj-mcmc Ireland estimates. The posterior estimates do show a reasonably large seasonal cycle, with emissions greatest during May–August and 20% greater than emissions between November–February. The largest contributor to Ireland's reported CH$_4$ emissions is the agriculture sector which account for 90% of reported national emissions. Similarly to the UK, the larger summertime emissions could therefore be representative of a summertime peak in agricultural emissions, or an indication of seasonal variation in natural wetland or grassland emissions.

## 6.2 Devolved administration emissions

The UK is composed of its four devolved administrations (DAs) of England, Scotland, Wales and Northern Ireland (NI), with a separate part of the NAEI prepared for each. We compare the rj-mcmc results for each DA to establish consistency with the DA inventories, and the degree to which these are independently resolved. Due to delays in the production of the DA inventories relative to the UK national inventory, we use the NAEI DA values from the 2020 report (NAEI2020).

Figure 5 shows the annual mean rj-mcmc emissions from each of the DAs alongside the corresponding NAEI2020 estimates as well as emissions estimates from the Republic of Ireland. NAEI2020 leaves a portion of emissions from the North Sea as unallocated to any DA, which we assign here to Scotland to be consistent with the inversion outputs. The posterior emissions estimates are consistent with NAEI2020 for each DA. We find the largest mean emissions from England of 1.48 (1.36–1.61) Tg yr$^{-1}$ between 2013–2018 compared to 1.42 Tg yr$^{-1}$ in NAEI2020 during the same period. The posterior estimates from England display a negative trend of -0.05 ±0.01 Tg yr$^{-2}$, accounting for the negative trend found in the total UK estimates.





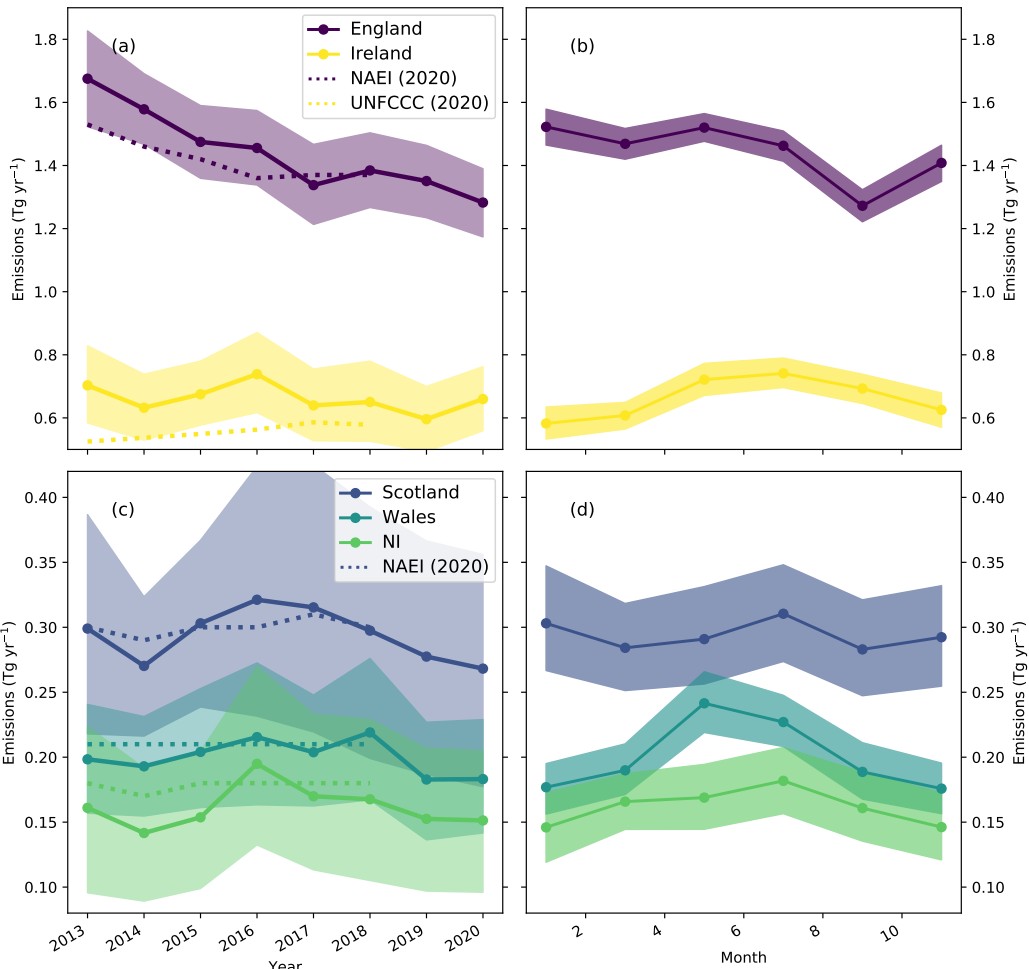

**Figure 5.** Annual emission estimates for the individual devolved administrations (DAs) of the UK and estimates for the Republic of Ireland. Showing: (a) the rj-mcmc annual mean estimates for England and Ireland; (b) the mean seasonal cycle for England and Ireland; (c) annual means for Scotland, Wales and Northern Ireland (NI); (d) the mean seasonal cycle for the respective DAs. The shading represents the 95% confidence interval. Dotted lines represent the DA NAEI estimates from the 2020 report.

Posterior rj-mcmc estimates for the other DAs are largely flat and consistent with no negative trend in NAEI2020. We find
small correlation coefficients for each 2-month inversion between the different DAs of between -0.05 to 0.08 indicating the posterior DA totals are independent of each other, and the atmospheric observation network provides an ability to independently resolve emissions from these sub-national regions of the order of $10^4$ to $10^5$ km$^2$.

Posterior 95% confidence intervals as a percentage of the annual means are of the order of 15–20% for England, 40–65% for Scotland, 40–50% for Wales and 70–80% for N. Ireland. We note that the annual mean uncertainty on the Scotland estimate
increased from 40% to 65% after 2015 following the decommissioning of the tall-tower measurement site in Angus, Scotland





**Table 2.** Comparison of posterior rj-mcmc emissions from different measurement networks in 2015. All units in Tg yr$^{-1}$ unless stated. DECC = BSD-HFD-MHD-RGL-TAC-TTA. GAUGE = DECC + GLA + Ferry

| Network | UK | UK trend (Tg yr$^{-2}$) | England | Scotland | Wales | NI | Ireland |
|---|---|---|---|---|---|---|---|
| NAEI (2020) | 2.13 | -0.03 | 1.42 | 0.30 | 0.21 | 0.18 | |
| Prior | 2.25 (1.33–3.38) | 0.0 | 1.52 (0.80–2.55) | 0.34 (0.16–0.57) | 0.20 (0.07–0.39) | 0.18 (0.06–0.32) | 0.62 (0.31–1.03) |
| DECC | 2.14 (2.06–2.21) | -0.05 ±0.02 | 1.47 (1.42–1.53) | 0.30 (0.27–0.34) | 0.20 (0.18–0.23) | 0.15 (0.13–0.18) | 0.67 (0.63–0.73) |
| RGL-TAC | 2.09 (1.97–2.21) | -0.06 ±0.04 | 1.43 (1.34–1.53) | 0.32 (0.27–0.38) | 0.18 (0.15–0.20) | 0.16 (0.12–0.20) | 0.58 (0.50–0.68) |
| MHD-TAC | 2.20 (2.08–2.32) | -0.07 ±0.04 | 1.47 (1.38–1.57) | 0.33 (0.28–0.40) | 0.21 (0.17–0.25) | 0.19 (0.15–0.23) | 0.76 (0.70–0.83) |
| TAC | 2.07 (1.93–2.22) | -0.04 ±0.05 | 1.39 (1.28–1.49) | 0.33 (0.27–0.41) | 0.19 (0.15–0.23) | 0.16 (0.12–0.20) | 0.61 (0.50–0.73) |
| MHD | 2.44 (2.13–3.11) | -0.03 ±0.06 | 1.65 (1.38–2.25) | 0.34 (0.27–0.43) | 0.24 (0.18–0.34) | 0.21 (0.17–0.26) | 0.78 (0.72–0.86) |
| Ferry | 1.92 (1.69–2.16) | NA | 1.25 (1.06–1.44) | 0.32 (0.26–0.40) | 0.19 (0.14–0.24) | 0.16 (0.12–0.23) | 0.57 (0.44–0.69) |
| FAAM (2014) | 2.19 (1.91–2.46) | NA | 1.43 (1.20–1.64) | 0.33 (0.23–0.45) | 0.16 (0.08–0.27) | 0.26 (0.14–0.40) | 0.46 (0.27–0.65) |
| GAUGE | 2.12 (2.05–2.19) | NA | 1.45 (1.40–1.50) | 0.32 (0.28–0.35) | 0.20 (0.18–0.22) | 0.15 (0.13–0.18) | 0.67 (0.62–0.72) |

(TTA). Four of the remaining five sites are located in England, which, along with the larger mean emissions, are the likely cause of the smaller uncertainties for England.

Figure 5 also shows the mean seasonal cycle of rj-mcmc emissions from each DA. Similar to Ireland we find a May–August peak in emissions from Wales and Northern Ireland that is 20–30% greater than winter emissions. The NAEI reports 70 and 80% of emissions from agriculture from Wales and Northern Ireland respectively, which could explain the summertime peak. Emissions from England and Scotland do not display a similar seasonal cycle, although we find a consistent dip in emissions from England during September–October.

## 6.3 Measurement network configuration comparison

In this section we investigate the impact that the volume and type of data used has on the UK rj-mcmc emissions estimates. The results presented in section 6.1 used all available tall-tower sites of the DECC network (although the exact number of stations varied from three to six, depending on the year and month). Here, we investigate how the UK means, uncertainties and trends are affected by the number of available measurement sites. We investigated how the emissions estimates are affected by using only one background site, MHD, or using one site regularly intercepting pollution peaks, TAC. We test the impact of using these two sites in combination, using two sites regularly intercepting pollution events (RGL and TAC) and through the use of two separate mobile measurement platforms.

Table 2 shows the 2015 annual mean posterior UK emissions that are estimated from these networks, together with the NAEI2020 and the 2013–2020 emissions trend. The results show the annual mean posterior uncertainties are over three times larger when using the single background site compared to using all available tower sites. However, using just TAC data leads to a 70% drop in the posterior uncertainty, with a slight further gain in combining the information of two measurement sites. The two site network of MHD-TAC can constrain annual UK CH$_4$ emissions to within a 95% confidence range of 0.24 Tg, compared to 0.15 Tg when using all available sites.



The 95% uncertainty range on the UK's NAEI reported total for 2018 is 0.68 Tg. On this basis, the MHD only inversion provides no uncertainty reduction, whereas both the two site network inversions provide at least a 50% reduction on this 95% range. Of course, the inversion accounts only for random uncertainties and likely underestimates the total uncertainty due to ignoring systematic errors. Even so, with only two measurement sites emissions are constrained to within a range of less than 15%. We find that a negative trend in UK $CH_4$ emissions is undetectable on an eight-year timescale in the MHD-only inversion, but is detected in both the 2-site inversions in addition to the DECC network inversion. At the regional level, we find correlation coefficients of less than 0.1 between most of the different DAs in both two site inversions. The 95% confidence range for England in 2015 decreases from 0.87 Tg for MHD only to 0.21 Tg for TAC only, 0.19 Tg for both two site inversions and 0.11 Tg for the full six sites. This shows that the TAC site on its own provides a large part of the constraint on England emissions. This is due to the site's sensitivity to regions of southern England that contain a large proportion of the UK's emissions (See Fig. 1(b)).

Figure 6 shows this spatial distribution of posterior uncertainty for some of the different networks. Low uncertainty values over most of southern and central England are seen when using TAC data alongside MHD, providing much greater constraint compared to just MHD. A similar feature is seen for Wales where most of the uncertainty reduction comes from the addition of one or both of RGL and TAC. For Scotland, most constraint comes from the TTA site in the DECC network. The results show, perhaps predictably, that measurement sites within a region (or downwind of emissions from a region) provide greatest uncertainty reduction of emissions within that region. The addition of the TAC site to MHD is enough to constrain both UK and England emissions to within a 95% confidence range of 0.25 Tg $yr^{-1}$. The additional sites of the DECC network increase confidence on regional emission estimates, but the annual means are similar and they add little additional constraint on the total UK estimate and trend due to the concentration of the majority of emissions in central and southern England.

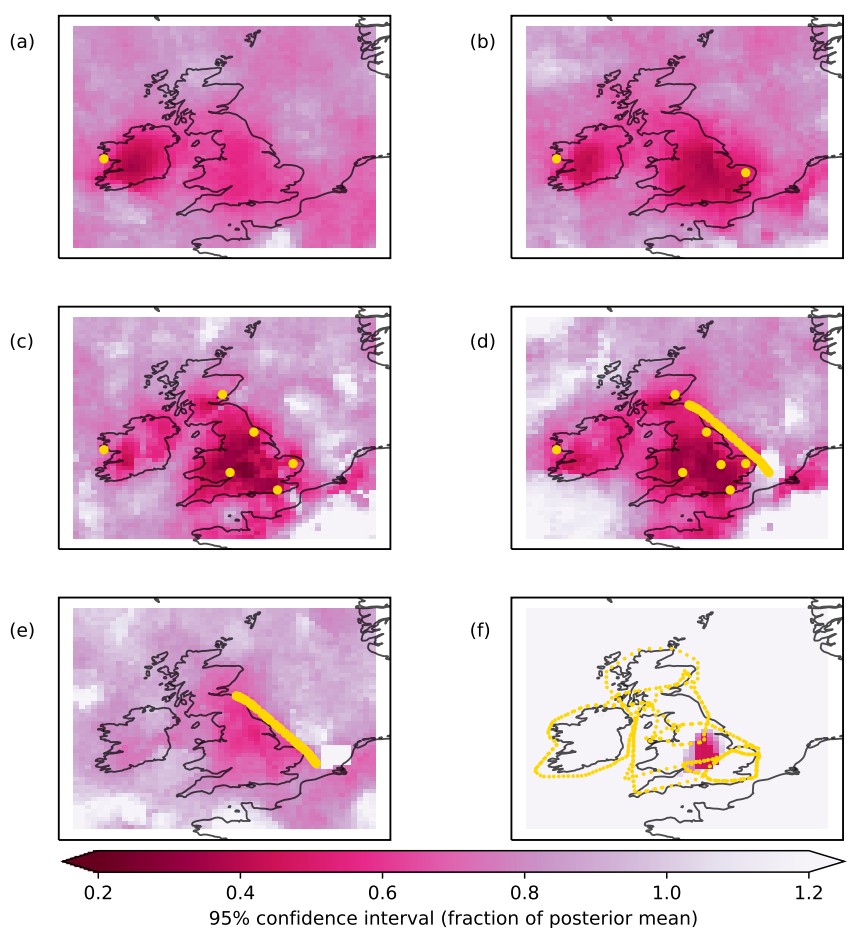

**Figure 6.** Posterior 95% confidence interval for 2015 as a fraction of the posterior estimate for each grid box. Darker colors indicate regions of lower uncertainty. Maps are shown for the following different measurement networks: (a) MHD, (b) MHD-TAC, (c) DECC network, (d) GAUGE network (DECC + GLA + Ferry), (e) Ferry, (f) FAAM aircraft (2014). Gold dots indicate locations of measurements used.

### 6.3.1 Mobile platforms and GAUGE network

Figure 3 and Table 2 also contain rj-mcmc estimates from the two mobile measurement sites. Both show a greater constraint on UK and DA emissions than the MHD only network. FAAM results are from 2014, covering seven flights, one in each of May and June, two in July and three in September 2014. The FAAM inversion was run using data averaged into five minute periods. As a sensitivity test, a smaller averaging time of 1 minute was tested and resulted in a UK estimate of 1.76 (1.52–2.00) Tg yr$^{-1}$ that was inconsistent with other networks. It is possible that data recorded at the time-scale of one minute was not



representative of the NAME model grid cells. The trade-off from averaging into periods of five minutes was a large reduction in data volume, and hence larger posterior uncertainties. Figure 6(f) shows the very limited area in which the posterior emissions
uncertainty is small when using this platform. To constrain national scale fluxes to the same degree as the two site networks on either two-monthly or annual time scales it would be necessary to perform a much larger number of flights than the seven used here in the absence of continuous surface data.

Posterior rj-mcmc emissions for 2015 derived from the ferry are slightly smaller than from the DECC network, primarily due to England emissions being smaller, but still overlap within the 95% confidence range. In addition, the UK mean over the
full sampling period of 2014–2016 was 2.02 (1.88–2.18) Tg yr$^{-1}$, more consistent with the DECC network mean over the same period of 2.16 (2.12–2.22) Tg yr$^{-1}$. Helfter et al. (2019) estimated a UK and Ireland emissions rate of 2.55 $\pm$0.48 Tg yr$^{-1}$ from the same ship-borne data between 2015–2017 using a mass balance approach. The estimate included Ireland due to the use of the MHD site as the measure of inflow for the mass balance calculations. Adding the Ireland component of our posterior emissions estimate to the UK gives 2.59 (2.37–2.84) Tg yr$^{-1}$, helping to reconcile these estimates. However, as shown in
Figure 6 the ship-borne posterior uncertainty estimates are lowest over Eastern parts of the UK and show little constraint over western parts and Ireland.

Finally, we combined all available data together in 2015, incorporating the six sites of the DECC tower network plus the ship-borne measurements and additional data from a church tower in Glatton, Cambridgeshire (GLA). We find similar UK CH$_4$ emissions compared to the DECC tower network of 2.12 (2.05–2.19) Tg yr$^{-1}$ in 2015, with no substantial changes in
emission estimates for any of the DAs or Ireland. Ostensibly, this can be explained by the additional data of GLA and the ferry providing constraint on regions such as southern England that are already well-sampled by the other measurement data. The value of the additional data is likely instead to lie in analysing smaller scale variations that are beyond the focus of this work.

### 6.4 Atmospheric transport model comparison

Figure 7 shows a comparison of UK and DA emissions estimated using NAME rj-mcmc and a second transport model, GEOS-
Chem, during 2015. The figure shows similar estimates of total UK emissions in each two month period, with estimates overlapping within the 95% confidence level. The 2015 mean UK estimate using GEOS-Chem was 2.08 (2.00–2.16) Tg yr$^{-1}$ compared to 2.14 (2.06–2.21) Tg yr$^{-1}$ from NAME. The two-monthly estimates overlap withn the range of the posterior uncertainties, with the greatest difference for emissions from England. There is a greater variability between the GEOS-Chem estimates of each two month period compared to NAME. In addition, the 95% confidence range of each two month estimate
from GEOS-Chem is 25% larger than from NAME, possibly due to a reduced ability to fit the data. The NAME inversions have a mean RMSE between the observations and modelled mole fractions of 10 ppb, compared to 22 ppb from the GEOS-Chem inversions.

Although the UK annual estimates are similar, the results show that the atmospheric transport model and inversion set up have a larger impact on the DA emissions and at sub-annual timescales. The GEOS-Chem inversion resolved only 14 independent
basis functions for UK emissions in each two month period, and 26 in total across the European inversion domain. The number of basis functions varied in the NAME rj-mcmc inversions and are difficult to isolate to the UK alone, but averaged 113 across



the inversion domain. Despite these differences the UK annual mean estimates are similar and both are slightly smaller than the 2.25 Tg assumed in the prior. A fuller comparison of the differences between transport models is beyond the scope of this work. However, our results suggest that the annual UK emissions estimated are not exclusive to the use of NAME. Improving

the spatial resolution of GEOS-Chem basis functions, by including a greater number of unknowns, may help to reconcile differences at the sub-national scale, and improve the fit of the GEOS-Chem posterior modelled mole fractions with the data.

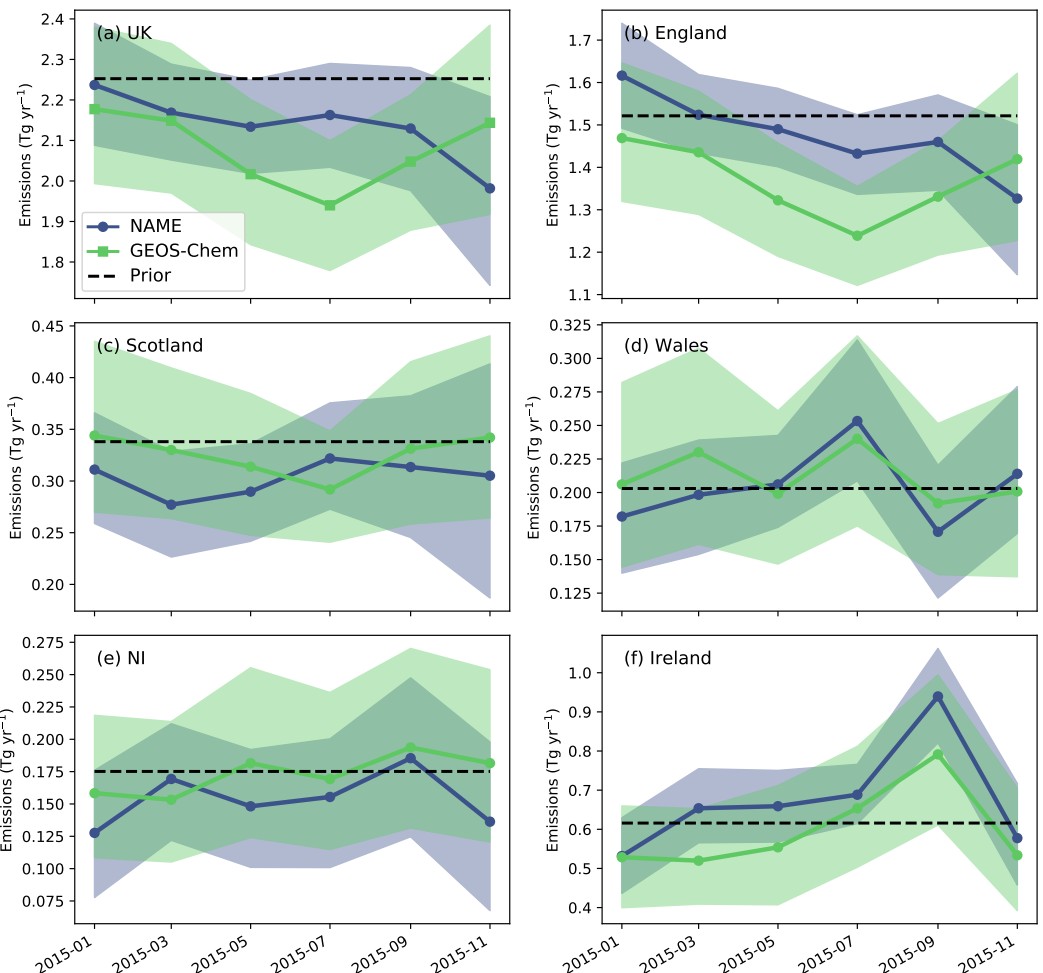

**Figure 7.** Two-monthly posterior estimates for 2015 from NAME (blue) and GEOS-Chem (green) inversions, and the prior used for the inversions (black dash). The panels show estimated emissions for (a) the UK, (b) England, (c) Scotland, (d) Wales, (e) NI and (f) Ireland. Shading represents the 95% confidence range of the inversion estimates.





## 6.5 Prior estimate comparison

The impact of the prior distribution of emissions was tested by using two further prior emission distributions. Firstly, we considered the case of using the EDGAR distribution of emissions over the UK instead of the NAEI. The EDGAR UK mean emissions are 30% larger than the NAEI at 2.92 Tg yr$^{-1}$. This allows us to investigate the impact of a potentially significant bias in the prior on the ability to maintain a consistent constraint on UK emissions. A second prior distribution assumed a flat rate of emissions throughout all land-based areas of Europe. For the purposes of this flat prior, emissions from the sea were considered to be negligible, and the annual mean UK emissions were 2.34 Tg yr$^{-1}$, although due to the relative areas of the different DAs, emissions from England were 23% smaller than the NAEI and almost 2.5 times larger from Scotland.

Figure 8 shows the annual mean emissions estimated for the UK and devolved administrations using these different priors, in addition to the main results. The annual mean estimates from both the EDGAR and flat prior inversions are larger than the main estimate, with limited to no overlap within the 95% confidence intervals. The annual mean emissions rates were 2.24 (2.15–2.33) Tg yr$^{-1}$ and 2.37 (2.25–2.49) Tg yr$^{-1}$ for the EDGAR and flat inversions respectively. The mean estimates of the EDGAR and flat inversions are 0.14 Tg and 0.27 Tg greater than our main results.

In both inversions, the main reason for the difference in UK emissions is due to differences in Scotland. Both EDGAR and flat priors were larger than the NAEI in Scotland by 0.08 Tg and 0.45 Tg respectively. This is reflected in the average annual mean posterior estimates of 0.36 (0.32–0.42) Tg yr$^{-1}$ and 0.53 (0.44–0.61) Tg yr$^{-1}$ from the respective inversions, compared to 0.29 (0.25–0.34) Tg yr$^{-1}$ from the main results. The flat prior inversion in particular maintains an offset in the posterior of 0.24 Tg that does not overlap within the range of the posterior uncertainties, indicating a lack of constraint on Scotland's emissions.

The spatial distribution of Scotland's emissions in the flat prior inversion is likely to be particularly unrealistic compared to other parts of the UK, due to the presence of substantial emissions in areas such as the sparsely populated Scottish highlands. Nevertheless, the results show that the atmospheric network is unable to correct for this likely error in the priors. This shortcoming of the measurement network is evident both before and after the decommissioning of the one measurement site (TTA) in Scotland, reflecting a lack of sensitivity to the northernmost part of the UK.

The results shown in Figure 8 demonstrate that the posterior emissions estimate from England is relatively robust to assumptions about the prior distribution and magnitude. All inversions estimate mean emissions for England of around 1.5 Tg yr$^{-1}$ and a negative trend of -0.05 Tg yr$^{-1}$ despite the prior means ranging from 77–140% of the NAEI value. England emissions account for around 70% of the UK total in the main results. The prior sensitivity tests demonstrate an independence on the distribution and magnitude of the prior for the majority of emissions.

A similar result is found for a two-site measurement network using measurements from only MHD and TAC (see Supplement). Emissions estimated for England from this two-site network and the NAEI prior are 1.47 (1.37–1.58) Tg yr$^{-1}$, compared to 1.44 (1.38–1.50) Tg yr$^{-1}$ from the full measurement network. The annual emissions estimates for England have a trend of -0.04 $\pm$0.02 Tg yr$^{-2}$. When using the EDGAR prior and just MHD-TAC data, a similar mean of 1.50 (1.38–1.61) Tg yr$^{-1}$ is found, and a trend of -0.04 $\pm$0.02 Tg yr$^{-2}$. With the flat prior, the mean emissions are 1.38 (1.28–1.48) Tg yr$^{-1}$, and the





trend -0.03 $\pm$0.02 Tg yr$^{-2}$. Although the results display larger differences than the full network, the majority of England's, and thus the UK's emissions, are relatively well-constrained by this two-site network and overlap within the 95% uncertainty range. However, greater differences between inversions using different priors are found for the other DAs when using only the two-site network, highlighting the importance of the denser measurement network for more robust evaluation of all the UK's

CH$_4$ emissions.

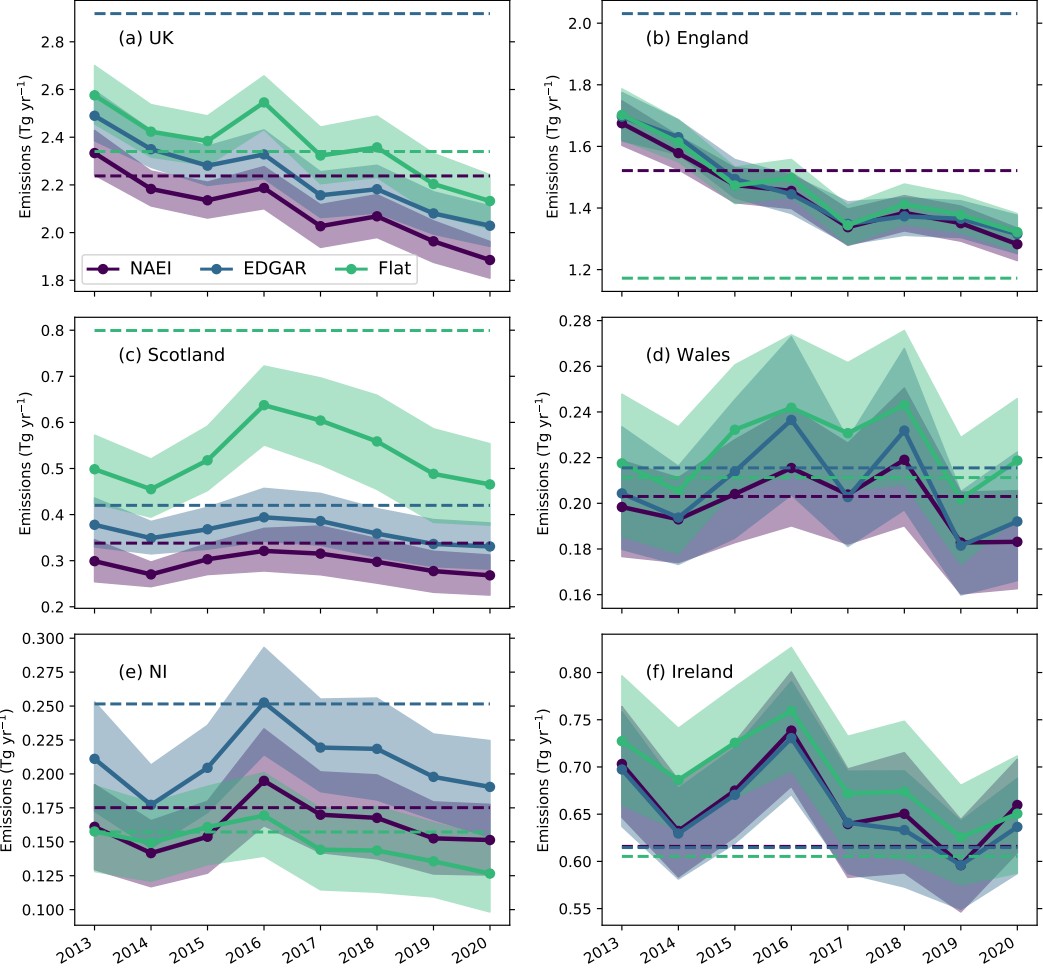

**Figure 8.** Annual mean emissions between 2013–2020 for inversions using different prior distributions from the NAEI (purple), EDGAR (blue) and a flat distribution (green). Estimates are shown for (a) the UK, (b) England, (c) Scotland, (d) Wales, (e) NI and (f) Ireland. Shading represents the 95% confidence interval. Dashed lines represent the magnitudes of the respective priors.



## 7 Discussion and conclusions

We show how the UK's $CH_4$ emissions can be evaluated from a network of tall-tower sites over the eight year period 2013–2020. Using a network of six measurement sites and a hierarchical Bayesian inversion method, emissions can be constrained to a 95% confidence interval that is within $\pm10\%$ of the mean value on a two-month timescale. A trend of -0.05 $\pm0.01$ Tg yr$^{-2}$ in the annual means is detectable by the network over the seven year period. We find a similar negative trend of -0.06 $\pm0.04$ Tg yr$^{-2}$ using a second inversion method (InTEM).

We show that similar results are achieved using a network of only two sites, with a 95% confidence interval of $\pm10\%$ and a trend of -0.03 Tg yr$^{-1}$. The results imply that, for constraining UK annual $CH_4$ emissions, a two-site network is sufficient, although this is somewhat dependent on the non-uniform distribution of emissions across the country. Although this work focuses on $CH_4$, this result also has relevance for the synthetic greenhouse gases such as hydrofluorocarbons. In the UK and Ireland these are measured only at MHD and TAC. Our results suggest that the addition of further instruments at other sites would not significantly change conclusions on the comparison with the NAEI for these gases, assuming the majority of emissions are within England.

At the level of devolved administrations, even when using the full measurement network, our results for Scotland are shown to be dependent on the prior emissions distribution. The current network has insufficient sensitivity to the northernmost parts of the UK. Our prior sensitivity tests show that the prior definition of emissions in this region carries over into the posterior estimates, with estimates not overlapping within the estimated uncertainties. This may not be overly significant for an assessment of anthropogenic $CH_4$, where sources in (sparsely populated) northern Scotland are thought to be minimal. However, for assessing natural sources such as wetland $CH_4$ or biospheric $CO_2$, this may be a more significant shortcoming.

The TTA site was decommissioned in late 2015 and there have been no measurements in Scotland since then as part of the UK's monitoring network. However, even when TTA data were available there was no convergence in the inversion estimates for Scotland from using different prior distributions. This is due to a lack of sensitivity to the northernmost parts of Scotland, and also the North Sea where there are significant oil and gas related emissions. To fully constrain emissions from the northernmost parts of the UK, measurements with greater sensitivity to both of these areas would be required. For England and Wales, we find the posterior estimates were not overly influenced by the magnitude or spatial distribution of the prior, as evidenced through sensitivity tests using the EDGAR distribution or a flat distribution of emissions.

The rj-mcmc inversion method solves for bulk $CH_4$ emissions, relying on the posterior scaling of the prior distribution to split the total $CH_4$ emissions into individual sectors. We find the higher emissions of summer time to be most likely due to the agriculture sector emissions, and the negative trend in annual emissions is most likely due to decreases in the waste and energy sectors. This second finding is in common with the NAEI, despite no trends being built into our priors. However, due to the lack of spatial independence in the prior distributions and the natural parsimony of the rj-mcmc inversion method we find positive correlations of 0.4–0.7 in our posterior sector estimates. Solving independently for the three main sector emissions could reduce this inter-dependence, although the same issue is likely to remain. Alternatively, the use of a co-tracer such as ethane, or direct measurements of the $\delta^{13}$C ratio may help to isolate emissions from the energy sector in particular.



Whilst this work has focused on an evaluation of the UK's methane emissions the results may have relevance for the process of national emissions verification as a whole. For a country the size of the UK ($\approx 2 \times 10^5$ km$^2$), a network of two in-situ measurement sites (sensitive to around 70% national total emissions) should be sufficient to constrain emissions at the national scale to within a 95% confidence range of around 10%. Additional measurement sites help reduce the posterior uncertainties on national and sub-national emissions, but their usefulness may ultimately lie in helping to address questions beyond the total

country annual emissions. Finally, we note that whilst the UK NAEI is prepared with a delay of two years, the atmospheric measurement emissions estimation can be carried out with a delay of at most a few months. This efficiency offers a potential advantage for using atmospheric measurements to track the UK's progress towards greenhouse gas reduction targets.

*Data availability.* Tower data from the UK DECC network are available via https://catalogue.ceda.ac.uk/uuid/f5b38d1654d84b03ba79060746541e4f. MHD data can be accessed from the AGAGE archive: https://agage.mit.edu/data/agage-data. Aircraft, ferry and GLA data from the GAUGE

project can be accessed via http://catalogue.ceda.ac.uk/uuid/9a1295858ff14fc6acea73e356a8842c.

*Author contributions.* MFL conducted the rj-mcmc inversions, GEOS-Chem simulations and wrote the paper. AJM conducted the NAME simulations and InTEM inversions. AJM and MR contributed to the writing of the paper. All other authors contributed data collection, calibration, analysis and quality control and provided comments on the paper.

*Competing interests.* The authors declare no competing interests.

*Acknowledgements.* University of Edinburgh authors were supported by Natural Environment Research Council (NERC) grants NE/K002449/1 and NE/S003819/1. University of Bristol authors were supported by the UK Department for Business, Energy and Industrial Strategy and NERC grants NE/I027282/1, NE/M014851/1, NE/L013088/1, NE/K002236/1, NE/S004211/1, NE/N016548/1 and NE/V00963X/1. University of Manchester authors were supported by NERC grants NE/K002449/1 and NE/K00221X/1. Since 2017, measurements at Heathfield have been maintained by the National Physical Laboratory mainly under funding from the National Measurement System.



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
