# Peer review of "Atmospheric observations consistent with reported decline in the UK's methane emissions, 2013 - 2020"

_Atmospheric Chemistry and Physics, 2021_

## Referee Comment (RC2)

**Summary**

The manuscript 'Atmospheric observations consistent with reported decline in the UK's methane emissions, 2013–2020' presented by Mark Lunt and Co-workers describes the results of atmospheric inversions of methane in the UK and Ireland. The manuscript builds on established inverse modelling frameworks but widens the scope of previous studies by applying these methods to a longer period, analysing the impact of variations in the observational network, the prior emissions and the employed transport model. As such the manuscript offers new insights and well-timed discussion on the requirements of observational networks for the estimation of nation emissions and their trends. The work is well organised and described and should be published after some minor corrections and additions (as detailed below) are implemented.

**Minor comments**

*Section 2.1*: The distinction between DECC and GAUGE is not always very clear. In the introduction they are described separately but in this section the title is just DECC. It would help to add a column to table 1 to indicate which network each station/platform is/was part of. A short statement of measurement uncertainty for the different sites/instruments would be good (maybe as part of Table1). How was this observation uncertainty considered in the inversions?

*Table 1*: It would also be good to include the full names of the sites in the table.

*Inversion domain*: It is not clear from the figures focusing on the British Isles how large the complete inversion domain was. It is hinted that European regions were part of the inversion, but were those only the ones shown for example in Figure 1? If the domain was as small as shown I would think that the boundary condition (baseline) problem may be much larger than currently discussed considering the large emissions located to the Southeast in the Benelux area. If the inversion domain was much larger the question arises, why no additional observations from other sites in Europe were considered?

*Section 3.2*: The description how source sensitivities were obtained with GEOS-Chem is rather brief. It would be good to provide a little more detail in order to better understand the limitations/advantages of the approach over the NAME approach.

*Section 4*: Outside the UK EDGAR v4.3.2 was used as prior distribution and again in the sensitivity tests. What was the reference year for EDGAR? There are also much newer versions of EDGAR available. Why not use them?

*L234*: Here it says that the 'model error correlation time scale' is part of the hierarchical approach, but on line 238f it says that it was fixed to 6 hours. What is correct?

*Model uncertainty*: rj-mcmc and InTem use quite different approaches for the model uncertainty used in the inverse step. Would it be possible to compare/comment on the finally used uncertainties? Could these be the main source

of the very different a posteriori uncertainties dervied by the systems?

*Section 5.3:* Is it correct to conclude that InTem used the less strict filter criterion for the observations? How much data was retained in the InTem case? Was there a seasonality in the amount of data used by the two systems and could that explain the differences in the seasonality of a posteriori emissions?

*Figure 3*: From the x-axis of the 2-monthly inversions it is not clear for which 2-month intervals the inversions were carried out. Jan/Feb, Mar/Apr, etc? Maybe this could be reflected by the axis labels instead of using just numbers.

*L300:* Please comment on which of the two uncertainty estimates (rj-mcmc vs. InTem) is more realistic. This is important when discussing the significance of any trend in the emissions.

*L302*: How were the trends and uncertainties calculated? Were the uncertainties of the posterior in the individual months taken into account? From Figure 3 I would have judged that the trend for InTEM is not significant given the large uncertainties.

*L339f*: The emission 'anomaly' in summer 2018 is much stronger in the rj-mcmc results than in InTem. Again the question if this could be due to different observations being assimilated. Furthermore, it would be interesting to see where these differences occur spatially.

*Figure 5 and related discussion*: Are there no InTem results available on the sub-national scale? It would be important to show that the inversions agree also on smaller than the national scale. In this regard, it would also be beneficial to include some figures of a posteriori distributions or increments in the manuscript or supplement. This concerns both the differences between the inversion systems but also the question how stable the distributions are over time.

*Section 6.3 and Table 2)*: From the description it is not clear what is really used in the last 3 cases. Ferry and FAAM is clear, different mobile platforms, but were these used in addition to the DECC towers or exclusively? Table 2 contains another case GAUGE what does this include? Later when the results are discussed these points become more clear, but the information should already be given when the cases are introduced. It would also be good to indicate in the table for which periods the inversions were run.

*Discussion of results in Section 6.3*: The impact of the network on national and sub-national totals is discussed, but what is the impact on the estimates by sector? If you conclude that UK total CH4 emissions can be estimated from just two sites this may be correct, but isn't there additional benefit if one could learn more on the emissions by sector? See additional comment below.

*Figure 6*: It is interesting to note that the posterior uncertainty on the continent seems to decrease most when just MHD or MHD/TAC are used in the inversion, whereas large parts of the Benelux area remain white when the whole network was used. How is that possible?

*L423f*: The comparison in Figure 3 is not against the MHD only inversions. So the second sentence only refers to Table 2. The 'Ferry' and 'FAAM' results in Figure 3 are never really commented on.

*L442f*: Were the flight observations neglected for the combined observations? Why?

*L449*: Figure 7 also includes results for Ireland not just UK and DA.

*L454*: Larger variability is not very obvious. Maybe for England, but certainly not for the other areas.

*L456*: Is this RMSE for all observations from all sites? Separating this for the different sites could indicate where GEOS-Chem has more trouble. Most likely for the more polluted sites due to a lack of resolution in the local transport. Furthermore, does a bias play a role in the RMSE calculation? If so it may be better to compared a bias-corrected (centered) RMSE.

*Section 6.5*: Also here it would be interesting to show and comment on the posterior distributions from these inversions. Where does the correction of the EDGAR prior happen? How much detail can be picked up from the flat prior?

*L514f*: I would be more careful with this conclusion. Halocarbons often have a much more pointy emission distribution than CH4. Furthermore, this distribution is much less well known both spatially but also in magnitude. For CH4 your conclusion may be correct but in the end differences between prior and posterior remained relatively small, even in the flat prior case. Hence, the small impact of additional sites may simply come from the high quality of the prior. You simply don't need more information to bring the emissions in agreement with the model.

*L541f*: Again, this conclusion is a bit general. The UK is an island! For similarly large countries/regions on the continent surrounded by other emitting ares, the conclusion may not be correct, because a separation of contributions from different countries may not be as straightforward for any inversion system.

*L544*: 'beyond the total country annual emissions'. Exactly this may however be the important information inversions should be able to deliver for example when evaluating if reduction measures in individual emission sectors or even processes where as successful as envisaged by mitigation measures. Really just a comment.

**Technical comments**

*Reference to supplementary material*: It would be nice to refer to a concrete figure/section/table of the supplement instead of just stating that something is available ....

*L42:* 'through' instead of 'though'.

*L90:* 'Tohoku University' instead of 'Tohoku university'.

*L203:* Add ',' after J.

*L357*: Delete 'the' before 'Ireland's'.

---

## Author Comment (AC1)

We thank the reviewers for their assessment of the work and constructive comments. We respond to each of their comments in turn below. For clarity, referee comments are designated by blue italics and our response is in black. Page and line numbers refer to the marked-up version of the manuscript.

**Referee 1**

*Authors use two different inverse modeling systems (although based on the same transport model) to estimate methane emissions and emission trends in UK during 2013–2020. They use data from a 6-tower network and other observations, to test the influence of the observing system configuration on the estimated emissions and find the emissions estimates and decreasing trends (in the better-constrained part of the domain, eg without Scotland) are stable against the changes in model setup and the observing network configuration. The ability to match inventory to within 10% and its changes with time with both inverse models is a notable achievement. The paper is well written and can be accepted, after applying technical corrections/minor revisions taking into account the reviewers' suggestions.*

*Detailed comments:*

*Line 156 Written as 'To ensure that the prior assumptions did not influence our derived emissions trend, our prior emissions did not vary with time' – This could be true only in case there is no impact of prior on final estimates, otherwise, no trend in the prior would lead to damping of the posterior trend. Just stating the prior emission trend is set to zero could be an alternative.*

Agreed, we have updated the text accordingly (see p.8, L.188).

*Line 180-200 [for the unfamiliar reader,] the description of rj-mcmc algorithm can be extended by adding few sentences of general introduction, explaining terms like region, parameter, hyper-parameter. Also, mentioning a need for applying Monte Carlo would be helpful (due to lognormal PDF?).*

We have extended the description of the rj-mcmc approach and the need for a Monte Carlo approach (see Section 5.1 p9-10 L196).

*Line 263 The uncertainty is set to '10% of a pollution event' (amplitude?). Is it about simulated or observed? Would be useful to discuss somewhere how well this uncertainty compares to the posterior mismatch and does this factor vary between sites?*

This should say "simulated" and has now been updated. Regarding the second half of this comment, in response to a similar comment from referee 2 we have included a further discussion of the different uncertainties from the rj-mcmc and InTEM inversions, the differences between the various sites, and the comparison to posterior mismatch as defined by the root mean square error. We have included an additional figure (new Figure 4) to reflect this, and a discussion is included in Section 6.1.2 (see p.17)

*Line 276-277 A reason for degraded performance under stable conditions over flat terrain could be the miscalculation of PBL height (related to surface temperature and nighttime surface heat balance) – not mentioned here.*

We agree these are plausible reasons for degraded performance and have added them to the text (see section 5.3 p12).

*Technical corrections:*

*Line 120 Should one read 'mole fractions' as 'simulated mole fractions'?*

Yes, we have updated the text as suggested.

*Line 547 Do authors mean 'greenhouse gas reduction' or 'greenhouse gas emissions reduction'?*

We have added "emissions" to this sentence.

**Referee 2**

*Summary The manuscript 'Atmospheric observations consistent with reported decline in the UK's methane emissions, 2013–2020' presented by Mark Lunt and coworkers describes the results of atmospheric inversions of methane in the UK and Ireland. The manuscript builds on established inverse modelling frameworks but widens the scope of previous studies by applying these methods to a longer period, analysing the impact of variations in the observational network, the prior emissions and the employed transport model. As such the manuscript offers new insights and well-timed discussion on the requirements of observational networks for the estimation of nation emissions and their trends. The work is well organised and described and should be published after some minor corrections and additions (as detailed below) are implemented.*

**Minor comments**

*Section 2.1 : The distinction between DECC and GAUGE is not always very clear. In the introduction they are described separately but in this section the title is just DECC. It would help to add a column to table 1 to indicate which network each station/platform is/was part of. A short statement of measurement uncertainty for the different sites/instruments would be good (maybe as part of Table1). How was this observation uncertainty considered in the inversions?*

We have attempted to clarify the distinction between the different networks at various points in the text. As suggested, we have added a column to Table 1 indicating the networks each site belongs to. We have further added the mean measurement uncertainty for each site from each inversion to a revised table 1 and a description of how these were calculated for each inversion method in section 5.1 and 5.2. For further discussion of uncertainties, we refer the reviewer to the results section, 6.1.2, where we have updated the text to discuss the posterior uncertainties from the rj-mcmc inversions and prescribed uncertainties of InTEM, and we have included an additional figure (new Figure 4) to illustrate these values (see changes p.4 l.95, p.17, section 6.1.2)

*Table 1 : It would also be good to include the full names of the sites in the table.*

Agreed, and updated.

*Inversion domain: It is not clear from the figures focusing on the British Isles how large the complete inversion domain was. It is hinted that European regions were part of the inversion, but were those only the ones shown for example in Figure 1? If the domain was as small as shown I would think that the boundary condition (baseline) problem may be much larger than currently discussed considering the large emissions located to the Southeast in the Benelux area. If the inversion domain was much larger the question arises, why no additional observations from other sites in Europe were considered?*

We acknowledge the maps and description of the inversion domain have left the impact of boundary conditions open to interpretation in the original submission. To be clear, both rj-mcmc and InTEM used an inversion domain much larger than is shown in the figures centred on the British Isles. This information was missing from the text, which we have corrected. We have also edited Figure 1 to show the full rj-mcmc spatially varying domain, covering NW Europe. The rj-mcmc domain also contained 6 fixed regions outside the spatially varying domain. The full NAME domain extended between -97.9 E to 39.4 E and 10.7 N to 79.1 N. The total InTEM domain is the same extent. No additional observations from Europe were included as we wanted to examine the capability of the UK's monitoring network on its own since these data are used in the verification work that forms part of the UK's annual inventory report (see new Fig. 1 and text updates P11. L243)

*Section 3.2 : The description how source sensitivities were obtained with GEOS-Chem is rather brief. It would be good to provide a little more detail in order to better understand the limitations/advantages of the approach over the NAME approach.*

We have now extended this description to give a full explanation and show the basis function definitions in an additional supplementary figure. (see Fig S1 and P7 L160)

*Section 4 : Outside the UK EDGAR v4.3.2 was used as prior distribution and again in the sensitivity tests. What was the reference year for EDGAR? There are also much newer versions of EDGAR available. Why not use them?*

We used 2010 as the reference year. Whilst newer versions and updated years up to 2018 are available we did not use these because we wanted to use a temporally-invariant prior to allow the data to be the main determinant in detecting any trend. The focus of the paper is on the UK emissions, which we show are relatively robust to the choice of prior emissions. The flat prior sensitivity test also adjusted the prior distribution in non-UK countries, and we show that the impact of this on UK emissions is limited in regions, such as England, that are well observed by the data.

*L234 : Here it says that the 'model error correlation time scale' is part of the hierarchical approach, but on line 238f it says that it was fixed to 6 hours. What is correct?*

Thank you for spotting this error. The correlation time scale was fixed to 6 hours, and we have corrected the text to reflect this. (P11, L259)

*Model uncertainty: rj-mcmc and InTem use quite different approaches for the model uncertainty used in the inverse step. Would it be possible to compare/comment on the finally used uncertainties? Could these be the main source 1 of the very different a posteriori uncertainties dervied by the systems?*

Yes, we can compare the final posterior uncertainties from rj-mcmc and those prescribed for InTEM. We have included a further figure (Figure 4) to show the mean uncertainties per site from the main DECC network inversions, along with the fit to the data (root mean square error) for each inversion. The posterior InTEM uncertainties are approximately 3 times larger than the posterior uncertainties from the rj-mcmc inversions. This is reflected in the posterior emission uncertainties which are approximately 3.5 times larger for the UK emission estimates. We have added a discussion of this in Section 6.1.2 (p17), which states:

*"Figure 4 shows the mean posterior estimates of the model-measurement uncertainty at each site from the rj-mcmc inversions and the prescribed uncertainties from the InTEM inversion. The results show that the rj-mcmc posterior model-measurement uncertainties are*

*on average around three times smaller than those used in the InTEM inversions. Figure 4 shows the rj-mcmc model-measurement uncertainties are more consistent with the posterior fit to the data at each site, as demonstrated through the root mean square error (RMSE) at each site. Therefore, the posterior emission uncertainties of the rj-mcmc inversion may be more representative of the emissions uncertainty, if uncertainties are dominated by non-systematic components. As a result, we concentrate the majority of our remaining analysis on the results of the rj-mcmc inversion rather than InTEM. The three times larger model-measurement uncertainty used in the InTEM inversions helps to explain the much larger posterior emission uncertainties, which are 3.5 times larger on average.*

*Posterior model-measurement uncertainties are smallest at those sites that are furthest removed from local sources. These include Bilsdale, Mace Head and Angus. Both Bilsdale and Angus inlets are over 200 m, whereas Mace Head is a background station. In contrast, the Ridge Hill, Tacolneston and Heathfield sites have lower measurement inlet heights, and are closer to large CH4 sources. These features are also reflected in the InTEM uncertainties, albeit with larger values."*

*Section 5.3: Is it correct to conclude that InTem used the less strict filter criterion for the observations? How much data was retained in the InTem case? Was there a seasonality in the amount of data used by the two systems and could that explain the differences in the seasonality of a posteriori emissions?*

Yes, the InTEM filtering criteria result in more observations being used (although with larger uncertainties as mentioned above). On average over all sites InTEM included 30% more observations than rj-mcmc. On average InTEM included 60% of available observations in the inversion whereas the stricter filter of rj-mcmc resulted in 45% of observations being used.

There is a seasonal cycle in terms of observations used in the rj-mcmc approach, with the maximum number of observations in summer (July-Aug) and lowest in winter (Nov-Dec). In contrast the InTEM selection results in more uniform coverage through the year. However, the rj-mcmc emissions seasonal cycle does not correlate with the seasonal cycle in observations, and the seasonal emissions are furthest from the prior at times when fewer observations are used. Conceptually this is opposite to expectations if the emissions seasonal cycle was influenced by seasonal changes in prior vs data influences.

We have added a plot of the number of observations used per 2-month period in the supplement and added a discussion of the influence of observation selection on the different seasonal cycles between InTEM and rj-mcmc to section 6.1.1.

*Figure 3 : From the x-axis of the 2-monthly inversions it is not clear for which 2-month intervals the inversions were carried out. Jan/Feb, Mar/Apr, etc? Maybe this could be reflected by the axis labels instead of using just numbers*

Yes it was Jan-Feb, Mar-Apr etc. We have updated the x-axis labels accordingly.

*L300: Please comment on which of the two uncertainty estimates (rj-mcmc vs. InTem) is more realistic. This is important when discussing the significance of any trend in the emissions.*

The posterior emission uncertainties reflect differences in the inversion setups. The rj-mcmc model-measurement uncertainties, which are derived in the inversion, are smaller than the same uncertainties in InTEM. This reflects the smaller RMSE between the model and data than would be assumed from the InTEM model-measurement uncertainty. Therefore, the posterior emission uncertainties of the rj-mcmc inversion may be more representative of the

emissions uncertainty, if uncertainties are dominated by non-systematic components. However, the InTEM approach prescribes higher model-measurement uncertainties to try and account for systematic model uncertainties. Hence, InTEM posterior emission uncertainties are larger.

We have added some additional discussion of the uncertainties and trends in section 6.1.2, referring to the new Figure 4. This shows the model-measurement uncertainties from the rj-mcmc and InTEM inversions, providing some greater context for the derived posterior emission uncertainties.

*L302 : How were the trends and uncertainties calculated? Were the uncertainties of the posterior in the individual months taken into account? From Figure 3 I would have judged that the trend for InTEM is not significant given the large uncertainties.*

Trends were calculated based on the annual mean values and uncertainties, and were calculated by least squares regression. The reviewer's judgment is correct that the calculated trend for InTEM is not significant, with a p-value of 0.2 based on a two-sided t-test. In contrast, the rj-mcmc trend has a p-value of 0.006 due to the smaller uncertainties. We have updated the text in Section 6.1 to include the p-values, the calculation of the trend and highlight that only the rj-mcmc results estimate a significant trend.

*L339f : The emission 'anomaly' in summer 2018 is much stronger in the rj-mcmc results than in InTem. Again the question if this could be due to different observations being assimilated. Furthermore, it would be interesting to see where these differences occur spatially.*

We have plotted the distribution of emissions in summer 2018 relative to the 2018 annual mean for each inversion. Both inversions show larger emissions over most of the UK particular the western half of mainland Britain. On average over all years, rj-mcmc tends to estimate higher summertime emissions. Taking this into account, relative to the seasonal averages of the respective inversions the anomalies of 2018 are the same at 0.2 Tg /yr. We have added this anomaly analysis to the text and the plot of 2018 emission distributions to the supplement (p16, L.380 and Fig S19).

*Figure 5 and related discussion: Are there no InTem results available on the sub-national scale? It would be important to show that the inversions agree also on smaller than the national scale. In this regard, it would also be beneficial to include some figures of a posteriori distributions or increments in the manuscript or supplement. This concerns both the differences between the inversion systems but also the question how stable the distributions are over time.*

We have included some further plots of emission distributions in the supplement to add useful information to address this. These show maps of the emissions from both inversions over the inversion period. We also include maps of the posterior-prior difference from the rj-mcmc inversions. We have added an analysis of InTEM results on the devolved administrations to section 6.2, including the InTEM results in Figure 6.  Results for England and Ireland are broadly consistent. For the remaining devolved administrations, the InTEM uncertainties are comparatively large, making any comparisons of differences redundant. As discussed above, the rj-mcmc model-measurement uncertainties are more consistent with the actual fit to the data, hence why we concentrate our analysis on these results.

*Section 6.3 and Table 2): From the description it is not clear what is really used in the last 3 cases. Ferry and FAAM is clear, different mobile platforms, but were these used in addition*

*to the DECC towers or exclusively? Table 2 contains another case GAUGE what does this include? Later when the results are discussed these points become more clear, but the information should already be given when the cases are introduced. It would also be good to indicate in the table for which periods the inversions were run.*

We have attempted to make the network definitions clearer in both the table, table caption and the text when they are first introduced (see updated Table 2).

*Discussion of results in Section 6.3 : The impact of the network on national and sub-national totals is discussed, but what is the impact on the estimates by sector? If you conclude that UK total CH4 emissions can be estimated from just two sites this may be correct, but isn't there additional benefit if one could learn more on the emissions by sector? See additional comment below.*

Our analysis per sector from the DECC network showed relatively high Pearson correlation coefficients between different sectors, as mentioned in the text. We find these correlations to be slightly larger in the case of the reduced 2-site measurement network of MHD-TAC. The values for MHD-TAC network are shown below (DECC network are in brackets):

Agriculture-Waste = 0.60 (0.60)

Agriculture-Energy = 0.49 (0.43)

Waste-Energy = 0.82 (0.73)

In the DECC network inversions we find statistically significant trends from the 2-monthly estimates for the waste and energy sectors ($p < 0.01$). However, for the reduced MHD-TAC network this is not the case ($p > 0.15$). We have added this information to the text to highlight the benefit of the multi-site network.

*Figure 6 : It is interesting to note that the posterior uncertainty on the continent seems to decrease most when just MHD or MHD/TAC are used in the inversion, whereas large parts of the Benelux area remain white when the whole network was used. How is that possible?*

This is due to a combination of the uncertainty being plotted as a proportion of the posterior mean scaling factor and the complication of different region definitions in the different rj-mcmc inversions. Because of the way basis functions move around during the inversion, in the inversion with fewer data, Voronoi regions may be fewer in number but the value within each Voronoi cell may appear more constrained over a larger area. When using more data sites, more Voronoi regions are used in these areas, increasing the number of boundaries between the discrete regions and thereby increasing the uncertainties. It highlights an issue with interpreting these uncertainty maps from the rj-mcmc inversions in regions that are further from the measurement stations. To avoid this confusion, we have reduced the plot to show only the UK and Ireland which are the focus of the work and where the emissions and Voronoi regions are generally well-constrained and interpretation of rj-mcmc uncertainties is easier to understand.

*L423f : The comparison in Figure 3 is not against the MHD only inversions. So the second sentence only refers to Table 2. The 'Ferry' and 'FAAM' results in Figure 3 are never really commented on.*

We think the first part is a misunderstanding from the juxtaposition of the two sentences. We have rewritten the opening of this paragraph to make our meaning clearer and added references in the text to the results shown in Fig 3.

*L442f : Were the flight observations neglected for the combined observations? Why?*

Yes, we omitted the aircraft observations. Given the limited temporal coverage of flight surveys (snapshots only), the focus was on consistent long-term monitoring sites, and there was insufficient overlap in the sampling period between GLA and the aircraft. We have added this discussion to the text to be more clear about what data has been used and what has not. (p 24, L 510).

*L449 : Figure 7 also includes results for Ireland not just UK and DA*

Corrected.

*L454 : Larger variability is not very obvious. Maybe for England, but certainly not for the other areas*

Agreed, we have deleted this line.

*L456 : Is this RMSE for all observations from all sites? Separating this for the different sites could indicate where GEOS-Chem has more trouble. Most likely for the more polluted sites due to a lack of resolution in the local transport. Furthermore, does a bias play a role in the RMSE calculation? If so it may be better to compared a bias-corrected (centered) RMSE.*

Yes, the RMSE was for all sites. It does not differ too much per site, with a range of bias-corrected RMSE of 19.6 ppb at Angus, Scotland to 22.9 ppb at Tacolneston, England. Essentially GEOS-Chem has more trouble than NAME representing mole fractions at all sites and this is not dependent on proximity to pollution sources. We have added this further information to Section 6.4.

*Section 6.5 : Also here it would be interesting to show and comment on the posterior distributions from these inversions. Where does the correction of the EDGAR prior happen? How much detail can be picked up from the flat prior?*

We have added a plot to the supplement showing the EDGAR and flat prior corrections as well as the difference between the priors and the NAEI. The results show corrections to EDGAR occur in England, predominantly from London and the Midlands. These are consistent with the differences between the NAEI and EDGAR prior. The flat prior results show a smooth emission field, indicating that little fine-grained information is picked up, although country averages are similar. This is likely in part due to the use of Voronoi cells in the rj-mcmc framework which are relatively uniform across the domain. This means that relatively coarse, smooth fields are resolved. Future work will explore how to introduce a more adaptive grid framework into the rj-mcmc approach to allow the data to explore regions of complexity where required by the data. Nevertheless, the smooth updates to the flat prior field are again consistent with the differences between the NAEI and flat prior distributions in the UK (see Fig S20 and Section 6.5).

*L514f : I would be more careful with this conclusion. Halocarbons often have a much more pointy emission distribution than CH4. Furthermore, this distribution is much less well known both spatially but also in magnitude. For CH4 your conclusion may be correct but in the end differences between prior and posterior remained relatively small, even in the flat prior case. Hence, the small impact of additional sites may simply come from the high quality of the prior. You simply don't need more information to bring the emissions in agreement with the model.*

This probably depends on the HFC in question. Some such as HFC-134a may be similarly distributed across the country due to their usage primarily in mobile AC (MAC) systems. We have added further caveats to state this may only hold for those with population-distributed emissions, such as from MAC and population-based refrigeration. Given the population distribution of the UK (weighted towards southern and central England), it is likely that TAC is sensitive to a large proportion of emissions of these HFCs (see P29, L587-592).

*L541f : Again, this conclusion is a bit general. The UK is an island! For similarly large countries/regions on the continent surrounded by other emitting ares, the conclusion may not be correct, because a separation of contributions from different countries may not be as straightforward for any inversion system.*

We have edited this part of the conclusions to comment only on the UK rather than generalising to other countries (see P.29 L614-620)

*L544 : 'beyond the total country annual emissions'. Exactly this may however be the important information inversions should be able to deliver for example when evaluating if reduction measures in individual emission sectors or even processes where as successful as envisaged by mitigation measures. Really just a comment.*

Indeed, and we plan to explore this more in future work. We've slightly rewritten this sentence to be clearer about our meaning.

***Technical comments***

*Reference to supplementary material: It would be nice to refer to a concrete figure/section/table of the supplement instead of just stating that something is available ....*

We have updated all references to supplementary material to be more specific.

*L42: 'through' instead of 'though'*

Corrected.

*L90: 'Tohoku University' instead of 'Tohoku university'.*

Corrected.

*L203: Add ',' after J.*

Corrected.

*L357 : Delete 'the' before 'Ireland's'.*

Corrected.